# LEARNING COOPERATIVE MEAN FIELD GAMES ON SPARSE CHUNG-LU GRAPHS

## ABSTRACT

Large agent networks are abundant in applications and nature and pose difficult challenges in the field of multi-agent reinforcement learning (MARL) due to their computational and theoretical complexity. While graphon mean field games and their extensions provide efficient learning algorithms for dense and moderately sparse agent networks, the case of realistic sparser graphs remains largely unsolved. Thus, we propose a novel cooperative mean field game (MFG) model based on the large class of Chung-Lu graphs including power law networks with coefficients above two. Besides a theoretical analysis, we design scalable learning algorithms which especially apply to the challenging class of graph sequences with finite first moment and infinite second moment. We compare our model and algorithms for various examples on synthetic and real world networks with MFG algorithms based on Lp graphons and graphexes. As it turns out, our approach outperforms existing methods in many examples and on various networks due to the special design aiming at an important, but so far hard to solve class of MARL problems.

## 1 INTRODUCTION

Despite the rapid developments in the field of multi-agent reinforcement learning (MARL) over the last years, systems with many agents remain hard to solve in general (Canese et al., 2021; Gronauer & Diepold, 2022). Mean field games (MFGs) (Caines et al., 2006; Lasry & Lions, 2007) are a promising way to model large agent problems in a computationally tractable way while providing a solid theoretical framework at the same time. The idea of MFGs is to abstract large, homogeneous crowds of small agents into a single probability distribution, the *mean field*. While MFGs have been used in various areas ranging from pedestrian flows (Achdou & Laurière, 2020) to oil production (Bauso et al., 2016), the assumption of indistinguishable agents is not fulfilled in many applications.

A particularly important class of MARL problems are those with many connected agents. Initially, these agent networks were modeled by combining the graph theoretical concept of graphons (Lovász, 2012) with MFGs, resulting in graphon MFGs (GMFGs) (Caines & Huang, 2019; 2021; Cui & Koeppl, 2022; Zhang et al., 2024). Since GMFGs only model often unrealistic dense graphs, subsequently MFG models based on Lp graphons (Borgs et al., 2018b; 2019) and graphexes (Veitch & Roy, 2015; Caron & Fox, 2017; Borgs et al., 2018a) were developed, called LPGMFGs and GXMFGs, respectively (Fabian et al., 2023; 2024). While these models facilitate learning algorithms in moderately sparse networks, they exclude sparser topologies. Formally, (LP)GMFGs and GXMFGs are designed exclusively for graphs with expected average degree going to infinity.

The learning literature contains various approaches to finding optimal behavior in MFGs, see Laurière et al. (2022a) for an overview. For example, Subramanian et al. (2022) develop a decentralized learning algorithm where agents are able to independently learn policies, while Guo et al. (2019; 2023) focus on Q-learning methods for general MFGs. For the case of cooperative MFGs without network interactions, also referred to as mean field control, various learning approaches exist (Ruthotto et al., 2020; Carmona et al., 2023; Gu et al., 2023). However, we are aware of only one work by Hu et al. (2023b) which learns policies for cooperative MFGs on dense networks, but not on sparse ones.

To learn policies for even sparser networks, we require a suitable graph theoretical framework, the well-known *Chung-Lu (CL) random graph model* (Aiello et al., 2000; 2001; Chung & Lu, 2002; 2006). The CL model aligns with our aim to model sparse, large agent networks because: (i) it can generate sparse networks, e.g. power laws, in a scalable way; (ii) it has a solid theoretical foundation

with convergence results; (iii) it possesses properties which are beneficial for the design of efficient approximate learning algorithms; (iv) it is conceptually simple despite its flexibility and rich structure. These points are explained and discussed in more detail in the next sections.

Leveraging CL graphs, we formulate the new class of Chung-Lu cooperative MFGs (CLCMFGs). CLCMFGs provide a theoretically well-motivated framework for learning agent behavior in challenging large networks where the average expected degree is finite, but the degree variance may diverge to infinity. On the algorithmic side, we provide a *two systems approximation* of CLCMFGs and corresponding learning algorithms to approximately learn optimal behaviour in these complex agent networks. Finally, we evaluate our novel CLCMFG learning approach for multiple problems on synthetic and real-world networks and compare it to different existing methods mentioned above. Overall, our contributions can be summarized as:

- We introduce CLCMFGs to model large cooperative agent populations on very sparse graphs with finite expected average degree:
- We give a rigorous theoretical analysis and motivation for CLCMFGs;
- We provide a two systems approximation and scalable learning algorithms for CLCMFGs;
- We show the capabilities of our CLCMFG learning approach on synthetic and real world networks for different examplary problems.

## 2 CHUNG-LU GRAPHS

The Chung-Lu random graph model (Aiello et al., 2000; 2001; Chung & Lu, 2002; 2006) provides an efficient way to generate large, sparse networks (Fasino et al., 2021). Compared to Lp graphons and graphexes, the CL framework can capture sparser, often more realistic graph structures illustrated by Figure 1. Next, we give a brief overview over CL graphs and point to Fasino et al. (2021) for details.

**Graph generation.** Suppose we want to generate a random graph with $N \in \mathbb{N}$ nodes. Then, in the CL model, first specify a weight vector $\boldsymbol{w} \in \mathbb{R}_+^N$ with one weight $w_i \in \mathbb{R}_+$ for each node $i \in \{1, \ldots, N\}$ and assume without loss of generality that weights are ordered such that $w_1 \leq w_2 \leq \ldots \leq w_N$. Intuitively, a node with high weight is more likely to have many connections than a node with small weight. Formally, two nodes $i$ and $j$ in the CL model are connected with probability $w_i \cdot w_j / \bar{w}$, independently of all other node pairs and with normalization factor $\bar{w} := \sum_{1 \leq k \leq N} w_k$. As discussed in Fasino et al. (2021), not all weight vectors yield valid probabilistic expressions $w_i \cdot w_j / \bar{w}$. Thus, for $N \in \mathbb{N}$ vertices we focus on the set of admissible weight vectors $\boldsymbol{\mathcal{W}}_N := \{\boldsymbol{w} \in \mathbb{R}_+^N : w_N^2 \leq \bar{w}\}$, unless stated otherwise. For given $N$, let $W_N$ be the weight $w_i$ of a uniformly at random chosen node $i \in \{1, \ldots, N\}$. Following Van Der Hofstad (2024, Chapter 1), the empirical distribution function of $W_N$ is $F_N(x) := \frac{1}{N} \sum_{i \leq N} \mathbf{1}_{\{w_i \leq x\}}$ for $x \geq 0$.

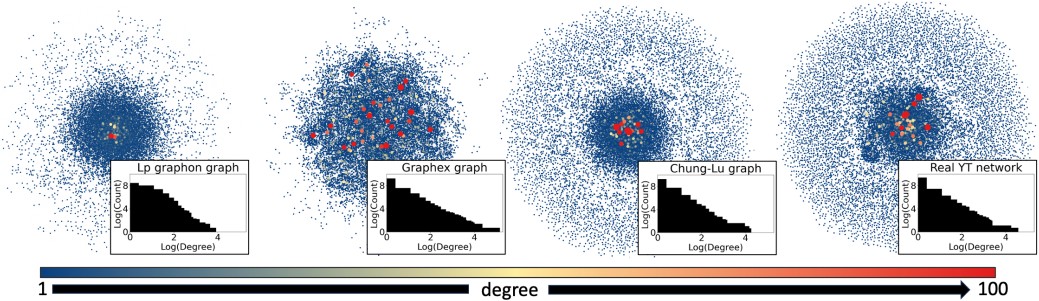

Figure 1: Four networks, first two generated by an Lp graphon and graphex, third is a CL graph and fourth is a real subsampled YouTube (YT) network (Mislove, 2009; Kunegis, 2013), highly connected nodes are depicted larger. Each network has around 14.5k nodes and 13k edges, except graphex has around 16.5k edges; all networks are plotted in the *prefuse force directed layout* (software: cytoscape). While the Lp graphon graph lacks sufficiently many high degree nodes, the tail of the graphex degree distribution is too heavy. In contrast, the CL graph is qualitatively close to the real YT network.

**Network convergence.**   For a meaningful theoretical analysis, the sequence of random CL graphs has to converge in a suitable sense given by the next assumption.

**Assumption 1** (Vertex weight convergence). *There exists a random variable $W$ with distribution function $F$ such that $\lim_{N \to \infty} F_N(x) = F(x)$ for any $x$ for which $x \mapsto F(x)$ is continuous. Furthermore, $\mathbb{E}[W_N] \to \mathbb{E}[W] \in (0, \infty)$ as $N \to \infty$.*

Assumption 1 ensures by $\lim_{N \to \infty} F_N(x) = F(x)$ that the weight $W_N$ of a uniformly at random picked node converges in distribution to the limiting random variable $W$ which is independent of $N$. Furthermore, $\mathbb{E}[W_N] \to \mathbb{E}[W] \in (0, \infty)$ states that the expectation of a randomly picked weight converges to the expectation of the limiting $W$. We emphasize the importance of $\mathbb{E}[W] \in (0, \infty)$ because it states a finite expected degree of randomly picked nodes, even as $N$ approaches infinity. The finite first moment is crucial for our approximate learning scheme introduced in the next sections and intuitively guarantees a relevant fraction of low degree agents in the limiting model.

We also tacitly assume $\operatorname{Var}(\deg(v_N)) \to \infty$ as $N \to \infty$ to ensure the existence of relatively many agents with (almost) infinitely many connections each. While we focus on the infinite variance case, our approach applies to the finite variance case as well. Since finite variance is easier to model by simply neglecting highly connected agents from our general approach, we focus on the challenging infinite variance case. Moving from vertex weight convergence in Assumption 1 to graph convergence requires a suitable graph convergence concept. We choose *local weak convergence in probability* which means that local node neighborhoods converge to neighborhoods in a limiting model. The next definition formalizes local weak convergence, for details see e.g. Lacker et al. (2023).

**Definition 1** (Local weak convergence in probability). *A sequence of finite graphs $(G_N)_N$ converges in probability in the local weak sense to $G$ if for all continuous and bounded functions $f : \mathcal{G}^* \to \mathbb{R}$*

$$\lim_{N \to \infty} \frac{1}{N} \sum_{i \in [N]} f(C_{v_i}(G_N)) = \mathbb{E}[f(G)] \qquad \text{in probability,}$$

*where $C_{v_i}(G_N)$ denotes the connected component of $v_i \in G_N$ with root $v_i$ and $\mathcal{G}^*$ is the set of isomorphism classes of connected rooted graphs.*

Under Assumption 1, large CL graphs exhibit a locally tree-like structure (Van Der Hofstad, 2024, Theorem 3.18) which is a key insight for the proofs of our theoretical results in the next sections. Throughout the paper, we use the running example of power law degree distributions with coefficient $\gamma > 2$ observed in many real world networks to some extent (Newman, 2003; Kaufmann & Zweig, 2009; Newman et al., 2011). However, our methods apply to all distributions meeting Assumption 1.

**Example 1** (Power law). *In our work, a power law is a zeta distribution with parameter $\gamma > 2$ such that $P(\deg(v) = k) = k^{-\gamma}/\zeta(\gamma)$, where $\zeta(\gamma)$ is the Riemann zeta function $\zeta(\gamma) := \sum_{j=1}^{\infty} j^{-\gamma}$.*

**Advantages.**   CL graphs are a flexible and efficient framework for generating large, sparse graph sequences. They have crucial advantages over other well-established graph generation approaches like the configuration model (CM) (Bender & Canfield, 1978; Wormald, 1980; Bollobás, 1998). Most notably, the CM generates multigraphs instead of simple graphs and the number of multiedges increases drastically as the vertex degrees increase. Consequently, the CM is suboptimal for generating graphs with a significant fraction of high degree nodes such as power law networks. Compared to the classical Barabási-Albert model (Barabási & Albert, 1999) which only generates power law networks with coefficient three (Bollobás et al., 2001), CL graphs are more flexible and can generate power law graphs with any coefficient larger than two as well as other network types. Besides its rich class of generable graphs, the CL model is theoretically well-founded and efficient implementations for large graph generation exist (Fasino et al., 2021). For a detailed discussion on the mentioned differences, see for example Chung & Lu (2002, Section 7).

Furthermore, as discussed in the introduction, the CL model can generate graphs with finite average expected degree which is a crucial advantage over both graphexes and (Lp) graphons. Finally, the neighbor degree distribution in CL graphs is described reasonably well by Heuristic 1 which provides the foundation for our two systems approximation. Both Heuristic 1 and the two systems approximation are defined and explained in detail in the following sections.

## 3 THE FINITE MODEL AND ITS LIMIT

In the following, we denote by $\mathcal{P}(\mathcal{X})$ the set of probability distributions over a finite set $\mathcal{X}$ and use the notation $[N] := \{1, \dots, N\}$ for any $N \in \mathbb{N}$.

**Finite model.** Assume some finite state space $\mathcal{X}$, finite action space $\mathcal{U}$ and finite and discrete time horizon $\mathcal{T} := \{0, \dots, T-1\}$ with terminal time point $T$ are given. Furthermore, there are $N \in \mathbb{N}$ agents connected by some graph $G_N = (V_N, E_N)$ with vertex set $V_N$ and edge set $E_N$. Here, the random state of agent $i \in [N]$ at time $t \in \mathcal{T}$ is denoted by $X_{i,t}^N$. All agents $V_N^k \subseteq V_N$ with degree $k \in \mathbb{N}$ share a common policy $\pi_t^k$ at all time points $t \in \mathcal{T}$. The empirical $k$-degree MF is defined as

$$\mu_t^{N,k} := \frac{1}{|V_N^k|} \sum_{i \in [N]: v_i \in V_N^k} \delta_{X_{i,t}^N} \in \mathcal{P}(\mathcal{X}),$$

for each time point $t \in \mathcal{T}$ and $k \in \mathbb{N}$. For notational brevity, define the overall empirical MF sequence as $\mu_t^N := (\mu_t^{N,1}, \mu_t^{N,2}, \dots) \in \mathcal{P}(\mathcal{X})^{\mathbb{N}}$. Each policy $\pi^k \in \mathcal{P}(\mathcal{U})^{\mathcal{T} \times \mathcal{X} \times \mathcal{G}^k}$ in the policy ensemble $\pi = (\pi^1, \pi^2, \dots) \in \mathcal{P}(\mathcal{U})^{\mathcal{T} \times \mathcal{X} \times \mathcal{G}^k \times \mathbb{N}}$ takes into account the current state of the respective agent $i$ with $k$ neighbors and its neighborhood $\mathbb{G}_{i,t}^N \in \mathcal{G}^k := \{G \in \mathcal{P}(\mathcal{X}) : k \cdot G \in \mathbb{N}_0^k\}$. Our learning algorithms also apply to other policy types, e.g., in our experiments we consider computationally efficient policies only depending on the current agent state. Then, the model dynamics are

$$U_{i,t}^N \sim \pi_t^k \left( \cdot | X_{i,t}^N, \mathbb{G}_{i,t}^N \right) \quad \text{and} \quad X_{i,t+1}^N \sim P \left( \cdot | X_{i,t}^N, U_{i,t}^N, \mathbb{G}_{i,t}^N \right),$$

for an agent $i$ with degree $k$, $t \in \mathcal{T}$, i.i.d. initial distribution $\mu_0 \in \mathcal{P}(\mathcal{X})$, and some transition kernel $P : \mathcal{X} \times \mathcal{U} \times \mathcal{P}(\mathcal{X}) \to \mathcal{P}(\mathcal{X})$. Note that the theory and subsequent learning algorithms extend to degree dependent transition kernels $P^k$. The policies are chosen to maximize the common objective

$$J^N(\pi) := \sum_{t=1}^{T} r(\mu_t^N)$$

with reward function $r : \mathcal{P}(\mathcal{X})^{\mathbb{N}} \mapsto \mathbb{R}$. Our model also covers reward functions with actions as inputs by using an extended state space $\mathcal{X} \cup (\mathcal{X} \times \mathcal{U})$ and splitting each time step $t \in \mathcal{T}$ into two.

**Limiting system.** In the limiting system, the MF for each degree $k \in \mathbb{N}$ evolves according to

$$\mu_{t+1}^k := \mu_t^k P_{t,\mu',W}^{\pi,k} := \sum_{\substack{(x,u) \in \mathcal{X} \times \mathcal{U}, \\ G \in \mathcal{G}^k}} \mu_t^k(x) P_{\pi} \left( \mathbb{G}_t^k (\mu_t') = G \mid x_t = x \right) \pi_t^k (u \mid x, G) P \left( \cdot \mid x, u, G \right)$$

with i.i.d. initial distribution $\mu_0^k \in \mathcal{P}(\mathcal{X})$ and where $\mathcal{G}^k$ is the set of $k$-neighborhood distributions as before. As in the finite system, define the limiting MF ensemble $\mu_t := (\mu_t^1, \mu_t^2, \dots) \in \mathcal{P}(\mathcal{X})^{\mathbb{N}}$ and the corresponding reward in the limiting system is $J(\pi) := \sum_{t=1}^{T} r(\mu_t)$.

**Theoretical results.** Next, we show the strong theoretical connection between the finite and limiting system. The following theoretical results built on the crucial observation (Van Der Hofstad, 2024, Theorem 3.18) that large CL graphs under Assumption 1 have a locally tree-like structure. Note that our theoretical results extend to arbitrary graph sequences converging in probability in the local weak sense. The proofs are in Appendix A. We first state empirical MF convergence to the limiting MFs.

**Theorem 1** (Mean field convergence). *Under Assumption 1 and for any fixed policy ensemble $\pi$, the empirical MFs converge to the limiting MFs such that for all $k \in \mathbb{N}$ and all $t \in \mathcal{T}$*

$$\mu_t^{N,k} \to \mu_t^k \quad \text{in probability for} \quad N \to \infty.$$

The MF convergence from Theorem 1 enables us to derive a corresponding convergence result for the finite and limiting objective functions under a standard continuity assumption on the reward.

**Assumption 2.** *The reward function $r : \mathcal{P}(\mathcal{X})^{\mathbb{N}} \mapsto \mathbb{R}$ is continuous.*

**Proposition 1** (Objective convergence). *Under Assumptions 1 and 2 and for any fixed policy ensemble $\pi$, the common objective in the finite system converges to the limiting objective, i.e.*

$$J^N(\pi) \to J(\pi) \quad \text{in probability for} \quad N \to \infty.$$

We leverage these findings to show that for a finite set of policy ensembles, the optimal policy for the limiting system in the set is also optimal in all sufficiently large finite systems. Therefore, if one wants to know the optimal ensemble policy for an arbitrary, large agent system, it suffices to find the optimal ensemble policy in the limiting system once which is formalized by Corollary 1.

**Corollary 1** (Optimal policy). *Assume some set $\{\pi_1, \ldots, \pi_M\}$ of $M < \infty$ policy ensembles is given and that w.l.o.g. $J(\pi_1) > J(\pi_i)$ for all $i \in [M]$ with $i \neq 1$. Under Assumptions 1 and 2 and for some $N^* \in \mathbb{N}$, $\pi_1$ is optimal in all finite systems of size $N > N^*$: $J^N(\pi_1) > \max_{i \in [M], i \neq 1} J^N(\pi_i)$.*

## 4 THE TWO SYSTEMS APPROXIMATION

In limiting systems on sparse graphs, the state evolution and optimal policy of an agent potentially depend on the entire network (Lacker & Soret, 2022). Calculating $P_{\boldsymbol{\pi}}\left(\mathbb{G}_t^k\left(\boldsymbol{\mu}_t'\right) = G \mid x_t = x\right)$ at time $t \in \mathcal{T}$ in the limiting system requires all possible $t$-hop neighborhood degree-state distributions where $t$-hop neighborhoods include all agents with a distance of at most $t$ edges to the initial agent. Unfortunately, Lemma 1 states that the number of $t$-hop neighborhoods grows at least exponentially with the agent degree $k$ in important classes of CL generated graphs, such as power laws beyond two.

**Lemma 1.** *In the limiting system, the number of possible $t$-hop degree-state neighborhood distributions of agents with degree $k \in \mathbb{N}$ at time $t \in \mathcal{T}$ in the worst case, e.g. power law, is $\Omega\left(2^{\mathrm{poly}(k)}\right)$.*

Just neglecting high degree nodes in the model might appear as a reasonable approximation to reduce computational complexity. However, the heavy tail of a degree distribution with finite expectation and infinite variance makes this approach highly inaccurate, as Example 2 illustrates.

**Example 2.** *In a power law graph with $\gamma = 2.5$ around $96\%$ of node degrees are at most five. These $96\%$ only account for roughly two thirds of the expected degree, formally $\sum_{h=1}^5 h^{1-\gamma}/\zeta(\gamma-1) < 0.68$. Nodes with a degree of at most ten still only account for around $76\%$ of the expected degree.*

**Two systems approximation.** For the subsequent two systems approximation, we first require a heuristic on the neighbor degree distribution for a given node.

**Heuristic 1.** *For an arbitrary node $v' \in V$ the degree distribution of its neighbor $v \in V$ is approximately $P(\deg(v) = k \mid \deg(v') = k', (v', v) \in E) \approx \frac{k \cdot P(\deg(v)=k)}{\sum_{k'' \in \mathbb{N}} k'' \cdot P(\deg(v)=k'')}$.*

Heuristic 1 is a good approximation for large CL graphs (Jackson et al., 2008, Chapter 4), and thus reasonable in our setup. As discussed in Jackson et al. (2008), Heuristic 1 is unrealistic for many other graph generators such as those using preferential attachment. The idea of Heuristic 1 is the following: if one fixes any node $v' \in V$ and considers its neighbors, high degree nodes are more likely to be connected to $v'$ than lowly connected ones. Instead of the overall degree distribution, we thus weight each probability by its degree and normalize accordingly. The result is an approximate neighbor degree distribution accounting for the increased probability of highly connected neighbors.

To address the complexity of the limiting system, we provide an approximate limiting system based on Heuristic 1 and the underlying CL graph structure. Our two systems approximation consists of a system for small degree agents with at most $k^*$ neighbors and another one for agents with more than $k^*$ connections, where $k^* \in \mathbb{N}$ is some arbitrary, but fixed finite threshold. Define an approximate MF $\hat{\mu}^k$ for each $k \in [k^*]$ and furthermore summarize all agents with more than $k^*$ connections into the infinite approximate MF $\hat{\mu}^\infty$ and define $\hat{\boldsymbol{\mu}} := (\hat{\mu}^1, \ldots, \hat{\mu}^{k^*}, \hat{\mu}^\infty)$. Based on Heuristic 1, we assume that all agents with more than $k^*$ neighbors observe the same neighborhood state distribution

$$\hat{\mathbb{G}}_t^\infty(\hat{\boldsymbol{\mu}}) := \frac{1}{\mathbb{E}[\deg(v)]}\left(\sum_{k=k^*+1}^\infty kP(\deg(v)=k)\right)\hat{\mu}_t^\infty + \frac{1}{\mathbb{E}[\deg(v)]}\sum_{h=1}^{k^*} hP(\deg(v)=h)\hat{\mu}_t^h.$$

The unified approximate neighborhood state distribution $\hat{\mathbb{G}}_t^\infty$ allows us to state an approximate, simplified version of the MF forward dynamics for high degree agents given by

$$\hat{\mu}_{t+1}^\infty := \hat{\mu}_t^\infty \hat{P}_{t,\boldsymbol{\mu}',W}^{\pi,\infty} := \sum_{x \in \mathcal{X}} \hat{\mu}_t^\infty(x) \sum_{u \in \mathcal{U}} \pi_t^\infty\left(u \mid x, \hat{\mathbb{G}}_t^\infty(\boldsymbol{\mu}')\right) P\left(\cdot \mid x, u, \hat{\mathbb{G}}_t^\infty(\boldsymbol{\mu}')\right),$$

where all agents with more than $k^*$ connections follow the same policy $\pi_t^\infty \in \mathcal{P}(\mathcal{U})^{\mathcal{T} \times \mathcal{X} \times \mathcal{P}(\mathcal{X})}$. The approximate neighborhood of an agent with degree $k \in [k^*]$ at each time $t \in \mathcal{T}$ is sampled from

$\hat{\mathbb{G}}_t^k(\hat{\boldsymbol{\mu}}) \sim \text{Mult}(k, \hat{\mathbb{G}}_t^\infty(\hat{\boldsymbol{\mu}}))$, i.e. $\hat{\mathbb{G}}_t^k(\hat{\boldsymbol{\mu}})$ is multinomial with $k$ trials and probabilities $\hat{\mathbb{G}}_t^\infty(\hat{\boldsymbol{\mu}})(x)$ for each $x \in \mathcal{X}$. Using Heuristic 1, the approximation yields for each $k \in [k^*]$ the MF forward dynamics

$$\hat{\mu}_{t+1}^k := \hat{\mu}_t^k \hat{P}_{t,\boldsymbol{\mu}',W}^{\pi,k} := \sum_{(x,u) \in \mathcal{X} \times \mathcal{U}} \sum_{G \in \boldsymbol{\mathcal{G}}^k} \hat{\mu}_t^k(x) P_{\text{Mult}}\left(\hat{\mathbb{G}}_t^k = G\right) \pi_t^k(u \mid x, G) P(\cdot \mid x, u, G) .$$

**Extensive approximation.** In Appendix B we derive a second, extensive approximation

$$P_{\boldsymbol{\pi},\boldsymbol{\mu}}\left(\mathbb{G}_{t+1}^k(\boldsymbol{\mu}_t) = G, x_{t+1} = x\right)$$

$$\approx \sum_{G' \in \boldsymbol{\mathcal{G}}^k} \sum_{x' \in \mathcal{X}} P_{\boldsymbol{\pi},\boldsymbol{\mu}}\left(\mathbb{G}_t^k(\boldsymbol{\mu}_t) = G', x_t = x'\right) \left[\sum_{u \in \mathcal{U}} \pi^k(u \mid x') P(x \mid x', u, G')\right]$$

$$\cdot \frac{\sum_{c \in \boldsymbol{\mathcal{C}}^k}\left[\sum_{\boldsymbol{a}_2 \in \boldsymbol{\mathcal{A}}_2^k(G',c)} \prod_j \text{Mult}_{\boldsymbol{p}_{2,j}}(\boldsymbol{a}_{2,j})\right] \sum_{\boldsymbol{a}_3 \in \boldsymbol{\mathcal{A}}_3^k(G,G',c)} \prod_{j,m} \text{Mult}_{\boldsymbol{p}_{3,jm}}(\boldsymbol{a}_{3,jm})}{\sum_{\boldsymbol{a}_2 \in \boldsymbol{\mathcal{A}}_2^k(G',c)} \prod_{j,m}\left(P(\deg(v) = m \mid (v',v) \in E) \mu_t^m(s_j)\right)^{a_{jm}}} .$$

of the finite agent neighborhoods. Here, the idea is to go beyond the previous multinomial assumption $\hat{\mathbb{G}}_t^k(\hat{\boldsymbol{\mu}}) \sim \text{Mult}(k, \hat{\mathbb{G}}_t^\infty(\hat{\boldsymbol{\mu}}))$ and to use state-degree neighborhood distributions $\boldsymbol{a}_2 \in \boldsymbol{\mathcal{A}}_2^k(G',c)$ and state-state-degree neighborhood distributions $\boldsymbol{a}_3 \in \boldsymbol{\mathcal{A}}_3^k(G,G',c)$ to capture agents changing from $x \in \mathcal{X}$ to $x' \in \mathcal{X}$ at a time step. We provide the extensive approximation derivation and corresponding definitions of sets like $\boldsymbol{\mathcal{A}}_2^k(G',c)$ and $\boldsymbol{\mathcal{A}}_3^k(G,G',c)$ in Appendix B. As we will see in the following, the extensive approximation often shows a moderately higher accuracy than our first approximation. However, the accuracy boost entails a significantly higher computational complexity due to multiple sums over sets like $\boldsymbol{\mathcal{A}}_2^k(G',c)$ and $\boldsymbol{\mathcal{A}}_3^k(G,G',c)$. Thus, our first approximation is more practical since it combines reasonable accuracy with low computational complexity.

## 5 LEARNING ALGORITHMS

To solve the MARL problem of finding optimal policies for each class of $k$-degree nodes, we propose two methods based on reducing the otherwise intractable many-agent graphical system to a single-agent MFC MDP. The first approach in Algorithm 1 is based on solving the resulting limiting MFC MDP under the parameters of the real graph, using the previously established two systems approximation. The second approach in Algorithm 2 instead directly learns according to single-agent RL that solves the MFC MDP by interacting with the real graph.

---

**Algorithm 1 Policy Gradient CLMFC**

1: **for** iterations $n = 1, 2, \ldots$ **do**
2:    **for** time steps $t = 0, \ldots, B_{\text{len}} - 1$ **do**
3:       Sample CLMFC MDP action $\boldsymbol{\pi}_t \sim \hat{\pi}^\theta(\boldsymbol{\pi}_t \mid \boldsymbol{\mu}_t)$.
4:       Compute reward $r(\boldsymbol{\mu}_t)$, next MF $\boldsymbol{\mu}_{t+1}$, termination flag $d_{t+1} \in \{0, 1\}$.
5:    **end for**
6:    Update policy $\hat{\pi}^\theta$ on minibatches $b \subseteq \{(\boldsymbol{\mu}_t, \boldsymbol{\pi}_t, r_t, d_{t+1}, \boldsymbol{\mu}_{t+1})\}_{t \geq 0}$ of length $b_{\text{len}}$.
7: **end for**

---

**RL in MFC MDP.** One can consider the two system approximation to reduce the complexity of otherwise intractable large interacting systems on networks to the MFs of each degree. The system state at any time is then given by low-degree MFs $\mu_t^1, \mu_t^2, \ldots, \mu_t^{k^*}$ and high-degree MF $\mu_t^\infty$, briefly $\boldsymbol{\mu}_t := (\mu_t^1, \mu_t^2, \ldots, \mu_t^{k^*}, \mu_t^\infty)$. Given a state $\boldsymbol{\mu}_t$, the possible state evolutions depend only on the analogous set of low-degree and high-degree policies at that time, $\boldsymbol{\pi}_t := (\pi_t^1, \pi_t^2, \ldots, \pi_t^{k^*}, \pi_t^\infty)$. In other words, choosing a particular $\boldsymbol{\pi}_t$ fully defines the state transition of the overall system, and can therefore be considered as the *high-level action* in the MFC MDP. Introducing a high-level policy $\hat{\pi}$ to output $\boldsymbol{\pi}_t \sim \hat{\pi}_t(\boldsymbol{\pi}_t \mid \boldsymbol{\mu}_t)$ allows us to solve for an optimal set of policies by solving the MFC MDP for optimal $\hat{\pi}$, since the MF dynamics are deterministic in the limit. Finally, the MFC MDP is solved by applying single-agent policy gradient RL, resulting in Algorithm 1. In practice, we use proximal policy optimization (Schulman et al., 2017). To lower the complexity of the resulting MDP, we parametrize policies as distributions over actions given the node state, $\pi_t^k \in \mathcal{P}(\mathcal{U})^{\mathcal{X}}$.

**MARL on real networks.** In addition to assuming knowledge of the model and computing the limiting MFC MDP equations, we may also directly learn on real network data without such model knowledge in a MARL manner. To do so, we still apply policy gradient RL to solve an assumed MFC MDP, but substitute samples from the real network into $\boldsymbol{\mu}_t$. At the same time, we let each node perform its actions according to the sampled $\boldsymbol{\pi}_t \sim \hat{\pi}_t(\boldsymbol{\pi}_t \mid \boldsymbol{\mu}_t)$. This approach is well justified by the previous theory and approximation, as for sufficiently large networks the limiting system and therefore also its limiting policy gradients are well approximated by this procedure.

---

**Algorithm 2 Policy Gradient CLMFMARL**

---

1: **for** iterations $n = 1, 2, \ldots$ **do**
2:     **for** time steps $t = 0, \ldots, B_{\mathrm{len}} - 1$ **do**
3:         Sample CLMFC MDP action $\boldsymbol{\pi}_t \sim \hat{\pi}^\theta(\boldsymbol{\pi}_t \mid \boldsymbol{\mu}_t)$.
4:         **for** node $i = 1, \ldots, N$ **do**
5:             Sample per-node action $U_{i,t} \sim \pi_t^{k_i}(U_{i,t} \mid X_{i,t})$ with degree $k_i = \infty$ if $k_i > k^*$.
6:         **end for**
7:         Perform actions, observe reward $r_t$, next MF $\boldsymbol{\mu}_{t+1}$, termination flag $d_{t+1} \in \{0, 1\}$.
8:     **end for**
9:     Update policy $\hat{\pi}^\theta$ on minibatches $b \subseteq \{(\boldsymbol{\mu}_t, \boldsymbol{\pi}_t, r_t, d_{t+1}, \boldsymbol{\mu}_{t+1})\}_{t \geq 0}$ of length $b_{\mathrm{len}}$.
10: **end for**

---

Overall, the approach results in Algorithm 2 and has a few advantages: Firstly, the algorithm does not assume model knowledge and is therefore a true MARL algorithm, in contrast to solving the limiting MFC MDP. Secondly, the algorithm avoids potential inaccuracies of the two systems approximation, as we will see in Section 7, since it directly interacts with a real network of interest. Lastly, in contrast to standard independent and joint learning MARL methods, the method is rigorously justified by single-agent RL theory and avoids exponential complexity in the number of agents respectively.

## 6 EXAMPLES

We consider four problems briefly described here. Problem details can be found in Appendix C.

**Susceptible-Infected-Susceptible/Recovered (SIS/SIR).** The classical SIS model (Kermack & McKendrick, 1927; Brauer, 2005) is a benchmark in the MFG learning literature (Laurière et al., 2022b; Zhou et al., 2024). Agents are infected or susceptible, resulting in the state space $\mathcal{X} \coloneqq \{S, I\}$, and decide to protect themselves or not. The infection probability increases without protection, and with the number of infected neighbors. The SIR model (Hethcote, 2000; Doncel et al., 2022) is an extension of SIS where agents can also be in an immune, recovered state $R$ such that $\mathcal{X} \coloneqq \{S, I, R\}$.

**Graph coloring (Color).** Inspired by graph coloring (Jensen & Toft, 2011; Barenboim & Elkin, 2022), the states are finitely many colors on a circle and a target color distribution is given. Agents stay at their color or costly move to a neighboring color. The objective decreases for deviations from the target color distribution and if neighbors of an agent have neighboring colors to the agent's color.

**Rumor.** In the rumor model (Maki & Thompson, 1973; Gani, 2000; Cui et al., 2022), agents are either aware of a rumor (aware state $A$) or they have not heard the rumor (ignorant state $I$). Aware agents decide whether they spread the rumor to their neighbors or not. They are awarded for spreading the rumor to unaware agents but loose reputation for telling the rumor to already aware agents.

## 7 SIMULATION & RESULTS

In this section, we numerically verify the two system approximation as well as the proposed learning algorithms by comparing them with baselines from the literature. The two systems approximation is compared with previous graph approximations such as graphex or Lp graphon MF equations, and the learning algorithms are verified against standard scalable independent learning methods such as IPPO (Tan, 1993; Papoudakis et al., 2021), due to the large scale of networks considered here. To generate

Table 1: Average expected total variation $\Delta\mu = \frac{1}{2T}\mathbb{E}\left[\sum_t\|\hat{\mu}_t - \mu_t\|_1\right] \in [0,1]$ of MF $\mu_t$ and empirical MF $\hat{\mu}_t = \sum_i \delta_{X_t^i}$ ($\pm$ std. dev., 50 trials), for the four models for four problems on eight real-world networks, CLCMFG* not displayed for last two problems since calculations exceed maximum runtime. Best result for each network-problem combination in bold.

| | Model | Average expected total variation $\Delta\mu$ in %, standard deviation in brackets | | | | | | | |
| | | CAIDA | Cities | Digg Friends | Enron | Flixster | Slashdot | Yahoo | YouTube |
|---|---|---|---|---|---|---|---|---|---|
| SIS | LPGMFG | 24.02 (1.25) | 28.16 (0.41) | 21.98 (0.26) | 24.77 (0.32) | 22.48 (0.07) | 23.70 (0.43) | 10.11 (2.10) | 22.94 (0.25) |
| | GXMFG | 9.07 (1.25) | 10.90 (0.41) | 4.72 (0.26) | 4.73 (0.32) | 3.78 (0.07) | 5.48 (0.43) | 9.31 (2.10) | 6.43 (0.25) |
| | CLCMFG | 2.59 (1.14) | 5.00 (0.40) | 3.57 (0.26) | 3.39 (0.31) | 1.60 (0.07) | 2.41 (0.43) | **3.59 (1.59)** | 3.53 (0.25) |
| | CLCMFG* | **1.75 (0.90)** | **4.20 (0.40)** | **3.02 (0.26)** | **2.67 (0.31)** | **0.90 (0.07)** | **1.70 (0.42)** | 3.81 (1.70) | **2.93 (0.25)** |
| SIR | LPGMFG | 9.11 (1.40) | 10.01 (0.34) | 8.68 (0.31) | 9.51 (0.32) | 8.99 (0.09) | 9.37 (0.38) | 4.88 (1.82) | 8.90 (0.23) |
| | GXMFG | 2.81 (1.10) | 2.63 (0.31) | 1.27 (0.29) | 0.99 (0.30) | 0.99 (0.09) | 1.58 (0.36) | 4.60 (1.71) | 1.79 (0.23) |
| | CLCMFG | 1.31 (0.87) | 1.36 (0.27) | 1.08 (0.28) | 0.91 (0.30) | 0.58 (0.08) | 0.99 (0.33) | **2.62 (1.30)** | 1.07 (0.23) |
| | CLCMFG* | **1.18 (0.82)** | **1.10 (0.26)** | **0.80 (0.27)** | **0.59 (0.28)** | **0.26 (0.08)** | **0.71 (0.29)** | 2.63 (1.30) | **0.78 (0.23)** |
| Color | LPGMFG | 38.73 (0.17) | 38.59 (0.09) | 38.70 (0.04) | 39.83 (0.06) | 39.55 (0.02) | 39.07 (0.06) | 34.18 (0.26) | 38.52 (0.04) |
| | GXMFG | 11.33 (0.13) | 7.90 (0.06) | 7.85 (0.02) | 4.91 (0.03) | 6.38 (0.01) | 6.81 (0.03) | 32.62 (0.24) | 8.76 (0.02) |
| | CLCMFG | **0.70 (0.12)** | **0.48 (0.05)** | **0.19 (0.02)** | **0.36 (0.04)** | **0.39 (0.02)** | **0.33 (0.04)** | **1.05 (0.19)** | **0.19 (0.03)** |
| Rumor | LPGMFG | 20.03 (2.15) | 22.56 (0.50) | 18.39 (0.55) | 20.27 (0.61) | 18.94 (0.16) | 19.70 (0.82) | 9.68 (3.76) | 19.23 (0.47) |
| | GXMFG | 6.98 (2.06) | 7.49 (0.49) | 3.33 (0.54) | 2.86 (0.58) | 2.65 (0.16) | 3.82 (0.79) | 9.01 (3.69) | 4.79 (0.47) |
| | CLCMFG | **3.06 (1.59)** | **4.31 (0.48)** | **3.00 (0.53)** | **2.62 (0.57)** | **1.73 (0.15)** | **2.41 (0.75)** | **5.01 (2.21)** | **3.27 (0.46)** |

artificial networks of different sizes we employ a CL-based graph sampling algorithm (Chung & Lu, 2002; Miller & Hagberg, 2011) from the Python NetworkX package.

We compare the accuracy of our model on different empirical datasets with Lp graphon and graphex based models and with our extensive approximation CLCMFG*, where computationally feasible, to see how much information is lost in the CLCMFG approximation. We use eight datasets from the KONECT database (Kunegis, 2013), where we substitute directed or weighted edges by simple undirected edges: CAIDA (Leskovec et al., 2007)($N \approx 26k$), Cities (Kunegis, 2013) ($N \approx 14k$), Digg Friends (Hogg & Lerman, 2012) ($N \approx 280k$), Enron (Klimt & Yang, 2004) ($N \approx 87k$), Flixster Zafarani & Liu (2009) ($N \approx 2.5mm$), Slashdot (Gómez et al., 2008) ($N \approx 50k$), Yahoo (Kunegis, 2013) ($N \approx 653k$), and YouTube (Mislove, 2009) ($N \approx 3.2mm$). See the references for details.

**Results.** First, we establish the usefulness of CLCMFGs and CLCMFG*s by comparing their dynamics to those of LPGMFGs and GXMFGs (Fabian et al., 2023; 2024) on eight real-world networks, see Figure 2 for examplary dynamics over time. As Table 1 shows, our CLCMFG approaches clearly outperforms the current LPGMFGs and GXMFGs methods for all empirical networks and problems. The extensive approximation CLCMFG* moderately outperforms CLCMFGs across datasets, except Yahoo. Since the extensive approximation is more detailed, it is often more accurate then the CLCMFG approximation. However, Table 1 lacks an evaluation for CLCMFG* on the Color and Rumor problem because the extensive approximation is computationally too expensive for these problems. Consequently, CLCMFG dynamics are the more practical choice since they are computationally tractable and yield a very reasonable performance across problems and datasets.

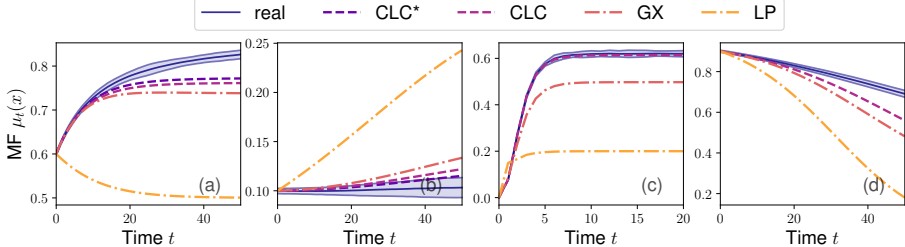

Figure 2: Overall MF evolution on real networks (50 trials, with two std. devs.), for our approx. (CLC), our extensive approx. (CLC*), graphex (GX), and Lp graphon (LP) models: (a) SIS on Enron, (b) SIR on Slashdot, (c) Color on CAIDA (without CLC*), (d) Rumor on Cities (without CLC*).

Table 2: (CL)MFC, (CL)MFMARL, and IPPO for four problems on synthetic graphs of size $N$. Best objective after 24 hours of training on 96 CPUs. Best result for each problem-graph tuple in bold.

| Problem | $N = 167$ | | | $N = 406$ | | | $N = 860$ | | | $N = 1598$ | | |
|---|---|---|---|---|---|---|---|---|---|---|---|---|
| | IPPO | MFC | MFMARL | IPPO | MFC | MFMARL | IPPO | MFC | MFMARL | IPPO | MFC | MFMARL |
| SIS | -20.80 | -14.56 | **-12.50** | -21.40 | -14.18 | **-11.64** | -19.70 | -12.42 | **-9.11** | -22.42 | -13.51 | **-11.13** |
| SIR | -7.45 | -7.84 | **-6.99** | -7.18 | -7.42 | **-6.55** | -10.64 | -6.86 | **-5.15** | -7.73 | -7.42 | **-6.32** |
| Color | -8.20 | -6.84 | **-6.74** | -8.05 | -7.04 | **-6.98** | -8.48 | -7.08 | **-5.85** | -8.15 | -6.97 | **-6.94** |
| Rumor | 0.24 | **1.19** | 0.27 | 0.16 | **1.33** | 0.19 | 0.25 | **1.47** | 1.35 | 0.12 | **1.33** | 0.17 |

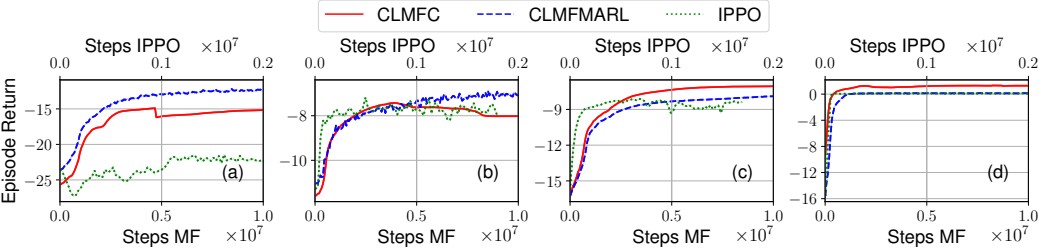

Figure 3: Training curves of CLMFC, CLMFMARL, and IPPO on a random CL graph with 406 nodes for: (a) SIS, (b) SIR, (c) Color, (d) Rumor.

The second part of our results focuses on our two learning algorithms CLMFC and CLMFMARL and compares them to the well-known IPPO algorithm. In Table 2, our algorithms outperform IPPO for all problems on the two larger graphs with 860 and 1598 nodes, respectively. On the two smaller graphs, CLMFC and CLMFMARL still yield an at least competitive performance compared to IPPO, where IPPO is only marginally better than CLMFC on two problem instances, namely SIR on $N = 167$ and $N = 406$. We point out that CLMFC, in contrast to IPPO and CLMFMARL, is not evaluated on the empirical system, but by design on the limiting CLCMFG model, which may differ from the true system behavior. These findings are complemented by the corresponding training curves in Figure 3. Finally, Figure 4 depicts how the training curves of our CLMFC and CLMFMARL algorithms converge on different empirical networks for different problems.

## 8 CONCLUSION

We have introduced the novel CLCMFGs which can depict agent networks with finite expected degree and diverging variance. After a theoretical analysis, we provided a practical two systems approximation which was then leveraged to design scalable learning algorithms. Finally, we evaluated the performance of our model and learning algorithms for different problems on synthetic and real-world datasets and compared them to existing methods. For future work, one could extend the CLCMFG model to various types of MFGs, e.g. to partial observability or agents under bounded rationality. We hope that CLCMFGs and the corresponding learning approach prove to be a versatile and useful tool for researchers across various applied research areas.

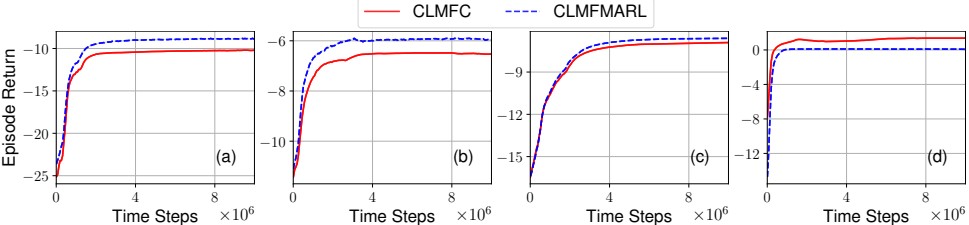

Figure 4: Training curves of CLMFC and CLMFMARL for four different examples: (a) SIS on Enron, (b) SIR on Slashdot, (c) Color on CAIDA, (d) Rumor on Cities.

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

# A    Appendix: Proofs for the Theoretical Results

## A.1    Proof of Theorem 1

*Proof.* We aim to eventually apply Lacker et al. (2023, Theorem 3.6) and therefore have to check that the respective conditions hold in our model. First, it is well-established (Van Der Hofstad, 2024, Theorem 3.18) that a Chung-Lu graph sequence $(G_N)_N$ under Assumption 1 converges in probability in the local weak sense to a unimodular branching process tree $G$ with offspring distribution

$$P(D = k) = \mathbb{E}\left[\exp(-W)\frac{W^k}{k!}\right].$$

Keeping in mind the i.i.d. initial distribution $\mu_0$, we leverage Lacker et al. (2023, Corollary 2.16) to obtain convergence in probability in the local weak sense of the marked graphs $(G_N, X^{G_N})$ to the limiting marked graph $(G, X^G)$.

Since the theory in Lacker et al. (2023) is only formulated in terms of particle systems without including actions in the form of policies, we provide a suitable reformulation of our cooperative mean field game model. Thus, define an auxiliary extended state space $\mathcal{X}_e := \mathcal{X} \cup (\mathcal{X} \times \mathcal{U})$ which serves as the state space for the extended particle system for some fixed policy ensemble $\pi$. The idea behind the extended state space $\mathcal{X}_e$ is to define an extended particle system where the state transition in $\mathcal{X}$ and the choice of the next action $u_{t+1} \in \mathcal{U}$ are separated into two different time steps.

Using the notations from Lacker et al. (2023), denote by $\mathcal{S}^{\sqcup}(\mathcal{X})$ the set of finite unordered sequences of arbitrary length with values in $\mathcal{X}$ and by $\Xi := \mathcal{X}^{\mathcal{X} \times \mathcal{S}^{\sqcup}(\mathcal{X})} \times \mathcal{U}^{\mathcal{X} \times \mathcal{U} \times \mathcal{S}^{\sqcup}(\mathcal{X})}$ the set of possible noise values. Next, specify a transition function $F^\tau : \mathcal{X}_e \times \mathcal{S}^{\sqcup}(\mathcal{X}_e) \times \Xi \to \mathcal{X}_e$ for each $\tau \in \mathcal{T}_e := \{0\} \cup [2T-1]$ by

$$X_{e,i,\tau+1}^N = F^\tau(X_{e,i,\tau}^N, \mathbb{G}_{e,i,\tau}^N, \xi_{i,\tau+1}) := \begin{cases} (X_{e,i,\tau}^N, \xi_{i,\tau+1}^0(X_{e,i,\tau}^N, \mathbb{G}_{e,i,\tau}^N)) & \text{if} \quad \tau/2 \in \{0\} \cup \mathbb{N} \\ \xi_{i,\tau+1}^1(X_{e,i,\tau}^N, \mathbb{G}_{e,i,\tau}^N) & \text{otherwise,} \end{cases}$$

where the neighborhood in the extended particle system $\mathbb{G}_{e,i,\tau}^N$ corresponds to $\mathbb{G}_{i,\lfloor \tau/2 \rfloor}^N$ in the original system. Here, the noise terms $\xi_{i,\tau+1} = (\xi_{i,\tau+1}^0, \xi_{i,\tau+1}^1)$ depict the used noise depending on whether $\tau$ is an even or odd number. If $\tau$ is an even number, i.e. $\tau/2 \in \{0\} \cup \mathbb{N}$, we use $\xi_{i,\tau+1}^0(X_{e,i,\tau}^N, \mathbb{G}_{e,i,\tau}^N)$ which is a $\mathcal{U}$-valued random variable with distribution

$$\xi_{i,\tau+1}^0(x, G) \sim \pi_{\tau/2}^k(\cdot \mid x, G)$$

for each neighborhood $G$ and state $x \in \mathcal{X}$, where $k$ is the degree of agent $i$. If $\tau$ is odd, i.e. we have $X_{e,i,\tau}^N \in \mathcal{X} \times \mathcal{U}$, we choose the first, $\mathcal{X}$-valued entry of $X_{e,i,\tau}^N$ as the $x$ in the above probability distribution.

The $\mathcal{X}$-valued noise component $\xi_{i,\tau+1}^1$ is distributed as follows: if $\tau$ is an odd number, the noise term is sampled from

$$\xi_{i,\tau+1}^1(x, u, G) \sim P(\cdot \mid x, u, G)$$

where $X_{e,i,\tau}^N = (x, u)$. If $\tau$ is even and thus $X_{e,i,\tau}^N \in \mathcal{X}$, we just choose some arbitrary, but fixed action $u' \in \mathcal{U}$ instead of $u$ in the above sampling process.

Now, it remains to check that Lacker et al. (2023, Assumption A) is satisfied by the extended particle system defined above. First, the noise terms $\xi_{i,\tau}$ are i.i.d. distributed for all agents $i \in [N]$ and with respect to all time points $\tau \in \mathcal{T}_e$ by construction. Finally, keeping in mind that the respective spaces are discrete, the map $F^\tau$ is continuous for each $\tau \in \mathcal{T}_e$. Therefore, Lacker et al. (2023, Theorem 3.6) yields the desired result.

$\square$

## A.2    Proof of Proposition 1

*Proof.* We want to show

$$J^N(\pi) \to J(\pi) \quad \text{in probability for} \quad N \to \infty$$

which is equivalent to

$$\sum_{t=1}^{T} r(\mu_t^N) \rightarrow \sum_{t=1}^{T} r(\mu_t) \quad \text{in probability for} \quad N \rightarrow \infty \,.$$

The reward $r$ is a continuous function by Assumption 2. Furthermore, by Theorem 1 we know that the empirical mean fields converge in probability to the limiting mean fields. Hence, we can apply the continuous mapping theorem (Mann & Wald, 1943; Van der Vaart, 2000) to obtain the desired result. $\qquad\square$

## A.3 Proof of Corollary 1

*Proof.* Quantify the gap $\Delta$ between the optimal and the second best solution as

$$\Delta := J(\pi_1) - \max_{i \in [M], i \neq 1} J(\pi_i) > 0 \,.$$

Keeping in mind Proposition 1, we know that the objectives of the finite systems eventually converge to the limiting mean field objectives as $N$ approaches infinity. Thus, there exists some $N^*$ such that

$$\max_{i \in [M]} |J^N(\pi_i) - J(\pi_i)| < \frac{\Delta}{2}$$

holds for all $N > N^*$. Finally, the above considerations allow us to bound the difference of interest

$$\begin{aligned}
&J^N(\pi_1) - \max_{i \in [M], i \neq 1} J^N(\pi_i) \\
&\quad = J^N(\pi_1) - J(\pi_1) + J(\pi_1) - \max_{i \in [M], i \neq 1} J^N(\pi_i) \\
&\quad = \underbrace{J^N(\pi_1) - J(\pi_1)}_{> -\Delta/2} + \underbrace{J(\pi_1) - \left( \max_{i \in [M], i \neq 1} J(\pi_i) \right)}_{= \Delta} + \left( \max_{i \in [M], i \neq 1} J(\pi_i) \right) - \max_{i \in [M], i \neq 1} J^N(\pi_i) \\
&\quad > \frac{\Delta}{2} + \min_{i \in [M], i \neq 1} J(\pi_i) - J^N(\pi_i) > \frac{\Delta}{2} - \frac{\Delta}{2} = 0,
\end{aligned}$$

for all $N > N^*$ which implies the desired statement

$$J^N(\pi_1) > \max_{i \in [M], i \neq 1} J^N(\pi_i)$$

and thereby concludes the proof. $\qquad\square$

## A.4 Proof of Lemma 1

*Proof.* Since we want to lower bound the number of possible $t$-hop neighborhoods $N_{G,t}$, we assume for simplicity that $t$-hop neighbors of the initial agent have at most degree $k$ themselves. Furthermore, we keep in mind the fact (Beck & Robins, 2007, Theorem 2.2) that in a $d$-dimensional simplex with edge length $\ell \in \mathbb{N}$, the number of integer points contained in the simplex is

$$\binom{d+\ell}{d} = \frac{(d+\ell)!}{d! \ell!} \,. \tag{1}$$

Since Lemma 1 considers the worst case, it suffices to prove the lower bound $\Omega\left(2^{\text{poly}(k)}\right)$ for one class of CL graphs. We choose our running example of power law graphs with coefficient above two and under Assumption 1. It is well known that large CL graphs under Assumption 1 are locally tree-like (Van Der Hofstad, 2024, Theorem 3.18) which we tacitly exploit in the following induction proof.

The proof is via induction over $t$. We start with $t = 1$ and the corresponding 1-hop neighborhood. The neighborhood consists of $k$ agents where each one has a degree in $[k]$ and a state in $\mathcal{X}$. Since the agents themselves are indistinguishable in our model, we focus on the degree-state neighborhood distributions. Then, the set of possible state-degree neighborhood distributions can be seen as the

integer points in a $(k + |\mathcal{X}| - 1)$-dimensional simplex with edge length $k$. Keeping in mind Equation (1) and the well-known Stirling approximation, see e.g. Marsaglia & Marsaglia (1990), we obtain

$$
\begin{aligned}
N_{G,t} &\geq \binom{k + |\mathcal{X}| - 1 + k}{k + |\mathcal{X}| - 1} \\
&= \frac{(2k + |\mathcal{X}| - 1)!}{(k + |\mathcal{X}| - 1)!k!} \\
&\overset{\text{Stirling}}{\sim} \sqrt{\frac{2\pi(2k + |\mathcal{X}| - 1)}{2\pi(k + |\mathcal{X}| - 1)2\pi k}} \frac{(2k + |\mathcal{X}| - 1)^{2k + |\mathcal{X}| - 1}}{(k + |\mathcal{X}| - 1)^{k + |\mathcal{X}| - 1}k^k} \\
&\geq \frac{1}{\sqrt{2\pi k}} \frac{(2k + |\mathcal{X}| - 1)^k}{k^k} \\
&\geq \frac{2^k}{\sqrt{2\pi k}} \\
&= 2^{k - 1/2 \cdot \log_2(2\pi k)} \in \Theta\left(2^{\text{poly}(k)}\right).
\end{aligned}
$$

Now, it remains to establish the induction step from $t$ to $t + 1$ where we assume that $N_{G,t} \in \Omega\left(2^{\text{poly}(k)}\right)$ holds. Then, instead of looking at the $(t + 1)$-hop neighborhoods of the initial agent, we can equivalently look at his or her 1-hop neighborhoods where each neighbor's 'extended state' now consists of the neighbor's $t$-hop neighborhood, where we ignore the edge between the neighbor and initial agent. Thus, the simplex edge length decreases by one from $k$ to $k - 1$ which is negligible for large $k$. Leveraging the induction assumption, we obtain

$$
\begin{aligned}
N_{G,t+1} &= \Omega\left(\binom{N_{G,t} + k}{k}\right) \\
&= \Omega\left(\frac{(N_{G,t} + k)!}{k! N_{G,t}!}\right) \\
&\overset{\text{Stirling}}{=} \Omega\left(\sqrt{\frac{N_{G,t} + k}{k N_{G,t}}} \frac{(N_{G,t} + k)^{N_{G,t} + k}}{N_{G,t}^{N_{G,t}} k^k}\right) \\
&= \Omega\left(\frac{1}{\sqrt{k}} \frac{(N_{G,t} + k)^k}{k^k}\right) \\
&\overset{\text{(IA)}}{=} \Omega\left(\frac{1}{\sqrt{k}} \frac{\left(2^{\text{poly}(k)}\right)^k}{k^k}\right) \\
&= \Omega\left(2^{k \cdot \text{poly}(k) - (k + 1/2)\log_2(k)}\right) \\
&= \Omega\left(2^{\text{poly}(k)}\right)
\end{aligned}
$$

which concludes the proof. $\qquad \square$

## B   EXTENSIVE APPROXIMATION DERIVATION

The goal of the following section is to establish a detailed approximation of the probability

$$
P_{\boldsymbol{\pi}, \boldsymbol{\mu}}\left(\mathbb{G}_{t+1}^k(\boldsymbol{\mu}_t) = G, x_{t+1} = x\right).
$$

First, we condition on the previous neighborhood distribution and the previous state of the agent at time $t$

$$
\begin{aligned}
&P_{\boldsymbol{\pi}, \boldsymbol{\mu}}\left(\mathbb{G}_{t+1}^k(\boldsymbol{\mu}_t) = G, x_{t+1} = x\right) \\
&\quad = \sum_{x' \in \mathcal{X}} P_{\boldsymbol{\pi}, \boldsymbol{\mu}}\left(\mathbb{G}_{t+1}^k(\boldsymbol{\mu}_t) = G, x_{t+1} = x, x_t = x'\right) \\
&\quad = \sum_{G' \in \mathcal{G}^k} \sum_{x' \in \mathcal{X}} P_{\boldsymbol{\pi}, \boldsymbol{\mu}}\left(\mathbb{G}_{t+1}^k(\boldsymbol{\mu}_t) = G, \mathbb{G}_t^k(\boldsymbol{\mu}_t) = G', x_{t+1} = x, x_t = x'\right).
\end{aligned}
$$

Now, we can decompose the above expression into three separate terms

$$P_{\boldsymbol{\pi},\boldsymbol{\mu}}\left(\mathbb{G}_{t+1}^k\left(\boldsymbol{\mu}_t\right) = G, \mathbb{G}_t^k\left(\boldsymbol{\mu}_t\right) = G', x_{t+1} = x, x_t = x'\right)$$
$$= \underbrace{P_{\boldsymbol{\pi},\boldsymbol{\mu}}\left(\mathbb{G}_t^k\left(\boldsymbol{\mu}_t\right) = G', x_t = x'\right)}_{(\mathrm{I})} \cdot \underbrace{P_{\boldsymbol{\pi},\boldsymbol{\mu}}\left(x_{t+1} = x \mid \mathbb{G}_t^k\left(\boldsymbol{\mu}_t\right) = G', x_t = x'\right)}_{(\mathrm{II})}$$
$$\cdot \underbrace{P_{\boldsymbol{\pi},\boldsymbol{\mu}}\left(\mathbb{G}_{t+1}^k\left(\boldsymbol{\mu}_t\right) = G \mid \mathbb{G}_t^k\left(\boldsymbol{\mu}_t\right) = G', x_{t+1} = x, x_t = x'\right)}_{(\mathrm{III})}$$

which allows us to handle each term individually. Since we require a recursive computation of the probability $P_{\boldsymbol{\pi},\boldsymbol{\mu}}\left(\mathbb{G}_{t+1}^k\left(\boldsymbol{\mu}_t\right) = G, x_{t+1} = x\right)$, the first term (I) will not be reformulated any further. The computation of the second term (II) is straight-forward, i.e.

$$P_{\boldsymbol{\pi},\boldsymbol{\mu}}\left(x_{t+1} = x \mid \mathbb{G}_t^k\left(\boldsymbol{\mu}_t\right) = G', x_t = x'\right) = \sum_{u \in \mathcal{U}} \pi^k\left(u \mid x'\right) P\left(x \mid x', u, G'\right).$$

Thus, it remains to approximate the third term (III)

$$P_{\boldsymbol{\pi},\boldsymbol{\mu}}\left(\mathbb{G}_{t+1}^k\left(\boldsymbol{\mu}_t\right) = G \mid \mathbb{G}_t^k\left(\boldsymbol{\mu}_t\right) = G', x_{t+1} = x, x_t = x'\right)$$
$$= P_{\boldsymbol{\pi},\boldsymbol{\mu}}\left(\mathbb{G}_{t+1}^k\left(\boldsymbol{\mu}_t\right) = G \mid \mathbb{G}_t^k\left(\boldsymbol{\mu}_t\right) = G', x_t = x'\right).$$

To ensure a reasonable approximation complexity, we make the simplifying assumption that the neighborhood distribution does not (crucially) depend on the current state of the agent of interest, i.e.

$$P_{\boldsymbol{\pi},\boldsymbol{\mu}}\left(\mathbb{G}_{t+1}^k\left(\boldsymbol{\mu}_t\right) = G \mid \mathbb{G}_t^k\left(\boldsymbol{\mu}_t\right) = G', x_t = x'\right) \approx P_{\boldsymbol{\pi},\boldsymbol{\mu}}\left(\mathbb{G}_{t+1}^k\left(\boldsymbol{\mu}_t\right) = G \mid \mathbb{G}_t^k\left(\boldsymbol{\mu}_t\right) = G'\right).$$

Thus, we focus on

$$P_{\boldsymbol{\pi},\boldsymbol{\mu}}\left(\mathbb{G}_{t+1}^k\left(\boldsymbol{\mu}_t\right) = G \mid \mathbb{G}_t^k\left(\boldsymbol{\mu}_t\right) = G'\right)$$

which requires an involved combinatorial argument to be calculated. The main difficulty in the calculation stems from the fact that the $k$ neighbors of the initial agent in general have different degrees, different states at time $t$ as well as different states at time $t + 1$. For notational convenience, we denote by $x_{1,t}, \ldots, x_{k,t}$ the states of the $k$ neighbors of the initial agent at time $t$ and by $\deg_1, \ldots, \deg_{k^*}, \deg_\infty \in \{1, \ldots, k\}$ the number of neighbors with the respective degree. Also, define $\mathcal{C}^k := \{c = (c_1, \ldots, c_{k^*}, c_\infty) \in \mathbb{N}_0^{k^*+1} : c_1 + \ldots + c_{k^*} + c_\infty = k\}$ for notational convenience. Then, the above probability can be expressed as

$$P_{\boldsymbol{\pi},\boldsymbol{\mu}}\left(\mathbb{G}_{t+1}^k\left(\boldsymbol{\mu}_t\right) = G \mid \mathbb{G}_t^k\left(\boldsymbol{\mu}_t\right) = G'\right)$$
$$= \sum_{c \in \mathcal{C}^k} P_{\boldsymbol{\pi},\boldsymbol{\mu}}\left(\mathbb{G}_{t+1}^k\left(\boldsymbol{\mu}_t\right) = G, \deg_1 = c_1, \ldots, \deg_{k^*} = c_{k^*}, \deg_\infty = c_\infty \mid \mathbb{G}_t^k\left(\boldsymbol{\mu}_t\right) = G'\right)$$
$$= \sum_{c \in \mathcal{C}^k} P_{\boldsymbol{\pi},\boldsymbol{\mu}}\left(\deg_1 = c_1, \ldots, \deg_{k^*} = c_{k^*}, \deg_\infty = c_\infty \mid \mathbb{G}_t^k\left(\boldsymbol{\mu}_t\right) = G'\right)$$
$$\cdot P_{\boldsymbol{\pi},\boldsymbol{\mu}}\left(\mathbb{G}_{t+1}^k\left(\boldsymbol{\mu}_t\right) = G \mid \deg_1 = c_1, \ldots, \deg_{k^*} = c_{k^*}, \deg_\infty = c_\infty, \mathbb{G}_t^k\left(\boldsymbol{\mu}_t\right) = G'\right).$$

In the remainder of the derivation, we will frequently use for all $s \in \mathcal{X}, m \in \mathbb{N}$, and $t \in \mathcal{T}$ the approximation

$$P\left(x_t^1 = s \mid \deg(v_1) = m, (v_0, v_1) \in E\right) \approx P\left(x_t^1 = s \mid \deg(v_1) = m\right) = \mu_t^m(s). \qquad (2)$$

Next, we make an auxiliary calculation to calculate the degree distribution of a (uniformly at random picked) node $v_1$ conditional on its state $x_t^1$ and that it is a neighbor of the initial node $v_0$ of interest

$$P\left(\deg(v_1) = m \mid x_t^1 = s, (v_0, v_1) \in E\right)$$
$$= \frac{P\left(\deg(v_1) = m \cap x_t^1 = s \mid (v_0, v_1) \in E\right)}{P\left(x_t^1 = s \mid (v_0, v_1) \in E\right)}$$
$$= \frac{P\left(\deg(v_1) = m \mid (v_0, v_1) \in E\right) P\left(x_t^1 = s \mid \deg(v_1) = m, (v_0, v_1) \in E\right)}{P\left(\deg(v_1) > k^* \cap x_t^1 = s \mid (v_0, v_1) \in E\right) + \sum_{k=1}^{k^*} P\left(\deg(v_1) = k \cap x_t^1 = s \mid (v_0, v_1) \in E\right)}$$

$$\overset{(2)}{\approx} \frac{P\left(\deg(v_1) = m \mid (v_0, v_1) \in E\right) \mu_t^m(s)}{P\left(\deg(v_1) > k^* \cap x_t^1 = s \mid (v_0, v_1) \in E\right) + \sum_{k=1}^{k^*} P\left(\deg(v_1) = k \cap x_t^1 = s \mid (v_0, v_1) \in E\right)}$$

$$= \frac{P\left(\deg(v_1) = m \mid (v_0, v_1) \in E\right) \mu_t^m(s)}{P\left(\deg(v_1) > k^* \mid (v_0, v_1) \in E\right) \mu_t^\infty(s) + \sum_{k=1}^{k^*} P\left(\deg(v_1) = k \mid (v_0, v_1) \in E\right) \mu_t^k(s)}$$

where we exploit that

$$P\left(\deg(v_1) > k^* \cap x_t^1 = s \mid (v_0, v_1) \in E\right) + \sum_{k=1}^{k^*} P\left(\deg(v_1) = k \cap x_t^1 = s \mid (v_0, v_1) \in E\right)$$

$$= P\left(\deg(v_1) > k^* \mid (v_0, v_1) \in E\right) P\left(x_t^1 = s \mid \deg(v_1) > k^*, (v_0, v_1) \in E\right)$$

$$+ \sum_{k=1}^{k^*} P\left(\deg(v_1) = k \mid (v_0, v_1) \in E\right) P\left(x_t^1 = s \mid \deg(v_1) = k, (v_0, v_1) \in E\right)$$

$$\overset{(2)}{\approx} P\left(\deg(v_1) > k^* \mid (v_0, v_1) \in E\right) \mu_t^\infty(s) + \sum_{k=1}^{k^*} P\left(\deg(v_1) = k \mid (v_0, v_1) \in E\right) \mu_t^k(s).$$

For the running example of power law degree distributions with exponent $\gamma \in (2, 3)$, the conditional degree distribution is approximately

$$P\left(\deg(v_1) = m \mid x_t^1 = s_j, (v_0, v_1) \in E\right)$$

$$\approx \frac{\frac{m^{1-\gamma}}{\zeta(\gamma-1)} \mu_t^m(s_j)}{\frac{1}{\zeta(\gamma-1)} \left[\sum_{\ell=k^*+1}^\infty \ell^{1-\gamma}\right] \mu_t^\infty(s_j) + \frac{1}{\zeta(\gamma-1)} \sum_{h=1}^{k^*} h^{1-\gamma} \mu_t^h(s_j)}$$

$$= \frac{m^{1-\gamma} \mu_t^m(s_j)}{\left[\sum_{\ell=k^*+1}^\infty \ell^{1-\gamma}\right] \mu_t^\infty(s_j) + \sum_{h=1}^{k^*} h^{1-\gamma} \mu_t^h(s_j)}.$$

Based on the above probability and by the symmetry of the model, we obtain

$$P_{\boldsymbol{\pi}, \boldsymbol{\mu}}\left(\deg_1 = c_1, \ldots, \deg_{k^*} = c_{k^*}, \deg_\infty = c_\infty \mid \mathbb{G}_t^k(\boldsymbol{\mu}_t) = G'\right)$$

$$= \sum_{\boldsymbol{a}_2 \in \mathcal{A}_2^k(G', c)} P_{\boldsymbol{\pi}, \boldsymbol{\mu}}\left(A_2 = \boldsymbol{a}_2, \deg_1 = c_1, \ldots, \deg_{k^*} = c_{k^*}, \deg_\infty = c_\infty \mid \mathbb{G}_t^k(\boldsymbol{\mu}_t) = G'\right)$$

$$= \sum_{\boldsymbol{a}_2 \in \mathcal{A}_2^k(G', c)} P_{\boldsymbol{\pi}, \boldsymbol{\mu}}\left(A_2 = \boldsymbol{a}_2 \mid \mathbb{G}_t^k(\boldsymbol{\mu}_t) = G'\right)$$

$$\approx \sum_{\boldsymbol{a}_2 \in \mathcal{A}_2^k(G', c)} \prod_{j=1}^d \binom{g_j'}{a_{j1}, \ldots, a_{j\infty}} \prod_{m \in [k^*] \cup \{\infty\}} \left(P\left(\deg(v_1) = m \mid x_t^1 = s_j, (v_0, v_1) \in E\right)\right)^{a_{jm}}$$

where we neglect dependencies between the nodes in the last line and define the matrix set $\mathcal{A}_2^k(G', c)$ for given $G' \in \boldsymbol{\mathcal{G}}^k$ and $c \in \boldsymbol{\mathcal{C}}^k$ as

$$\mathcal{A}_2^k(G', c) := \left\{\boldsymbol{a}_2 = (a_{jm})_{j \in [d], m \in [k^*] \cup \{\infty\}} \in \mathbb{N}_0^{d \times (k^*+1)} :\right.$$

$$\left. \sum_{m' \in [k^*] \cup \{\infty\}} a_{jm'} = g_j', \forall j \in [d] \quad \text{and} \quad \sum_{\ell=1}^d a_{\ell m} = c_m, \forall m \in [k^*] \cup \{\infty\}\right\}.$$

Therefore, it remains to calculate the conditional probability

$$P_{\boldsymbol{\pi}, \boldsymbol{\mu}}\left(\mathbb{G}_{t+1}^k(\boldsymbol{\mu}_t) = G \mid \deg_1 = c_1, \ldots, \deg_{k^*} = c_{k^*}, \deg_\infty = c_\infty, \mathbb{G}_t^k(\boldsymbol{\mu}_t) = G'\right).$$

As a first step, we define the set of matrices $\mathcal{A}_3^k(G, G', c)$ for a given triple of vectors $G, G' \in \boldsymbol{\mathcal{G}}^k$ and $c \in \boldsymbol{\mathcal{C}}^k$ as

$$\mathcal{A}_3^k(G, G', c) := \left\{\boldsymbol{a}_3 = (a_{ijm})_{i, j \in [d], m \in [k^*] \cup \{\infty\}} \in \mathbb{N}_0^{d \times d \times (k^*+1)} :\right.$$

$$\sum_{m' \in [k^*] \cup \{\infty\}} \sum_{\ell=1}^{d} a_{i\ell m'} = g_i \text{ and } \sum_{m' \in [k^*] \cup \{\infty\}} \sum_{\ell=1}^{d} a_{\ell j m'} = g_j', \quad \forall i, j \in [d]$$

$$\left. \text{and } \sum_{\ell,\ell'=1}^{d} a_{\ell\ell'm} = c_m, \forall m \in [k^*] \cup \{\infty\} \right\}.$$

where $d := |\mathcal{X}|$ is the finite number of states. Intuitively, the matrix set $\mathcal{A}_3^k(G, G', c)$ for an agent with degree $k$ contains all possible numbers $(a_{ijm})_{i,j \in [d], m \in [k^*] \cup \{\infty\}}$ of neighbors whose degree is $m$ and current state is $x_i$ and who transition to state $x_j$ in the next time step. For notational convenience, let $A$ denote the random variable taking values in $\mathcal{A}_3^k(G, G', c)$ and analogously let $A_2$ be the random variable with values in $\mathcal{A}_2^k(G', c)$. We continue with the reformulation

$$P_{\boldsymbol{\pi},\boldsymbol{\mu}} \left( \mathbb{G}_{t+1}^k(\boldsymbol{\mu}_t) = G \mid \deg_1 = c_1, \ldots, \deg_{k^*} = c_{k^*}, \deg_\infty = c_\infty, \mathbb{G}_t^k(\boldsymbol{\mu}_t) = G' \right)$$

$$= \sum_{\boldsymbol{a}_2 \in \mathcal{A}_2^k(G',c)} P_{\boldsymbol{\pi},\boldsymbol{\mu}} \left( A_2 = \boldsymbol{a}_2 \mid \deg_1 = c_1, \ldots, \deg_{k^*} = c_{k^*}, \deg_\infty = c_\infty, \mathbb{G}_t^k(\boldsymbol{\mu}_t) = G' \right)$$

$$\cdot P_{\boldsymbol{\pi},\boldsymbol{\mu}} \left( \mathbb{G}_{t+1}^k(\boldsymbol{\mu}_t) = G \mid A_2 = \boldsymbol{a}_2, \deg_1 = c_1, \ldots, \deg_{k^*} = c_{k^*}, \deg_\infty = c_\infty, \mathbb{G}_t^k(\boldsymbol{\mu}_t) = G' \right)$$

$$= \sum_{\boldsymbol{a}_2 \in \mathcal{A}_2^k(G',c)} P_{\boldsymbol{\pi},\boldsymbol{\mu}} \left( A_2 = \boldsymbol{a}_2 \mid \deg_1 = c_1, \ldots, \deg_{k^*} = c_{k^*}, \deg_\infty = c_\infty, \mathbb{G}_t^k(\boldsymbol{\mu}_t) = G' \right)$$

$$\cdot P_{\boldsymbol{\pi},\boldsymbol{\mu}} \left( \mathbb{G}_{t+1}^k(\boldsymbol{\mu}_t) = G \mid A_2 = \boldsymbol{a}_2 \right).$$

Next, we consider the two conditional probabilities separately. We start with

$$P_{\boldsymbol{\pi},\boldsymbol{\mu}} \left( A_2 = \boldsymbol{a}_2 \mid \deg_1 = c_1, \ldots, \deg_{k^*} = c_{k^*}, \deg_\infty = c_\infty, \mathbb{G}_t^k(\boldsymbol{\mu}_t) = G' \right)$$

$$= \frac{P_{\boldsymbol{\pi},\boldsymbol{\mu}} \left( A_2 = \boldsymbol{a}_2 \cap \deg_1 = c_1, \ldots, \deg_{k^*} = c_{k^*}, \deg_\infty = c_\infty, \mathbb{G}_t^k(\boldsymbol{\mu}_t) = G' \right)}{P_{\boldsymbol{\pi},\boldsymbol{\mu}} \left( \deg_1 = c_1, \ldots, \deg_{k^*} = c_{k^*}, \deg_\infty = c_\infty, \mathbb{G}_t^k(\boldsymbol{\mu}_t) = G' \right)}$$

$$= \frac{P_{\boldsymbol{\pi},\boldsymbol{\mu}} \left( A_2 = \boldsymbol{a}_2 \right)}{P_{\boldsymbol{\pi},\boldsymbol{\mu}} \left( \deg_1 = c_1, \ldots, \deg_{k^*} = c_{k^*}, \deg_\infty = c_\infty, \mathbb{G}_t^k(\boldsymbol{\mu}_t) = G' \right)}.$$

Keeping in mind both

$$P_{\boldsymbol{\pi},\boldsymbol{\mu}} \left( A_2 = \boldsymbol{a}_2 = (a_{jm})_{j,m} \right) \approx \prod_{j=1}^{d} \prod_{m \in [k^*] \cup \{\infty\}} \left( P\left( \deg(v_1) = m \mid (v_0, v_1) \in E \right) \mu_t^m(s_j) \right)^{a_{jm}}$$

by neglecting dependencies between the nodes and

$$P_{\boldsymbol{\pi},\boldsymbol{\mu}} \left( \deg_1 = c_1, \ldots, \deg_{k^*} = c_{k^*}, \deg_\infty = c_\infty, \mathbb{G}_t^k(\boldsymbol{\mu}_t) = G' \right)$$

$$= \sum_{\boldsymbol{a}_2 \in \mathcal{A}_2^k(G',c)} P_{\boldsymbol{\pi},\boldsymbol{\mu}} \left( \deg_1 = c_1, \ldots, \deg_{k^*} = c_{k^*}, \deg_\infty = c_\infty, \mathbb{G}_t^k(\boldsymbol{\mu}_t) = G', A_2 = \boldsymbol{a}_2 \right)$$

$$= \sum_{\boldsymbol{a}_2 \in \mathcal{A}_2^k(G',c)} P_{\boldsymbol{\pi},\boldsymbol{\mu}} \left( A_2 = \boldsymbol{a}_2 \right)$$

$$= \sum_{\boldsymbol{a}_2 \in \mathcal{A}_2^k(G',c)} \prod_{j=1}^{d} \prod_{m \in [k^*] \cup \{\infty\}} \left( P\left( \deg(v_1) = m \mid (v_0, v_1) \in E \right) \mu_t^m(s_j) \right)^{a_{jm}}$$

we obtain

$$P_{\boldsymbol{\pi},\boldsymbol{\mu}} \left( A_2 = \boldsymbol{a}_2 \mid \deg_1 = c_1, \ldots, \deg_{k^*} = c_{k^*}, \deg_\infty = c_\infty, \mathbb{G}_t^k(\boldsymbol{\mu}_t) = G' \right)$$

$$= \frac{P_{\boldsymbol{\pi},\boldsymbol{\mu}} \left( A_2 = \boldsymbol{a}_2 \right)}{P_{\boldsymbol{\pi},\boldsymbol{\mu}} \left( \deg_1 = c_1, \ldots, \deg_{k^*} = c_{k^*}, \deg_\infty = c_\infty, \mathbb{G}_t^k(\boldsymbol{\mu}_t) = G' \right)}$$

$$\approx \frac{\prod_{j=1}^{d} \prod_{m \in [k^*] \cup \{\infty\}} \left( P\left( \deg(v_1) = m \mid (v_0, v_1) \in E \right) \mu_t^m(s_j) \right)^{a_{jm}}}{\sum_{\boldsymbol{a}_2' \in \mathcal{A}_2^k(G',c)} \prod_{j=1}^{d} \prod_{m \in [k^*] \cup \{\infty\}} \left( P\left( \deg(v_1) = m \mid (v_0, v_1) \in E \right) \mu_t^m(s_j) \right)^{a_{jm}'}}$$

and especially, for the case of a power law degree distribution with $\gamma \in (2,3)$, we have

$$P_{\boldsymbol{\pi},\boldsymbol{\mu}}\left(A_2 = \boldsymbol{a}_2 \mid \deg_1 = c_1, \ldots, \deg_{k^*} = c_{k^*}, \deg_\infty = c_\infty, \mathbb{G}_t^k\left(\boldsymbol{\mu}_t\right) = G'\right)$$

$$= \frac{\prod_{j=1}^d \left(\mu_t^\infty(s_j)\left(1 - \sum_{m'=1}^{k^*} \frac{(m')^{1-\gamma}}{\zeta(\gamma-1)}\right)\right)^{a_{j\infty}} \prod_{m=1}^{k^*} \left(\frac{m^{1-\gamma}\mu_t^m(s_j)}{\zeta(\gamma-1)}\right)^{a_{jm}}}{\sum_{\boldsymbol{a}_2' \in \mathcal{A}_2^k(G',c)} \prod_{j=1}^d \left(\mu_t^\infty(s_j)\left(1 - \sum_{m'=1}^{k^*} \frac{(m')^{1-\gamma}}{\zeta(\gamma-1)}\right)\right)^{a_{j\infty}'} \prod_{m=1}^{k^*} \left(\frac{m^{1-\gamma}\mu_t^m(s_j)}{\zeta(\gamma-1)}\right)^{a_{jm}'}}$$

$$\approx \frac{\prod_{j=1}^d \left(\mu_t^\infty(s_j)\left(\zeta(\gamma-1) - \sum_{m'=1}^{k^*}(m')^{1-\gamma}\right)\right)^{a_{j\infty}} \prod_{m=1}^{k^*} \left(m^{1-\gamma}\mu_t^m(s_j)\right)^{a_{jm}}}{\sum_{\boldsymbol{a}_2' \in \mathcal{A}_2^k(G',c)} \prod_{j=1}^d \left(\mu_t^\infty(s_j)\left(\zeta(\gamma-1) - \sum_{m'=1}^{k^*}(m')^{1-\gamma}\right)\right)^{a_{j\infty}'} \prod_{m=1}^{k^*} \left(m^{1-\gamma}\mu_t^m(s_j)\right)^{a_{jm}'}}.$$

Now, it remains to calculate the second probability term, namely

$$P_{\boldsymbol{\pi},\boldsymbol{\mu}}\left(\mathbb{G}_{t+1}^k\left(\boldsymbol{\mu}_t\right) = G \mid A_2 = \boldsymbol{a}_2\right).$$

Exploiting the symmetry of the problem, we obtain

$$P_{\boldsymbol{\pi},\boldsymbol{\mu}}\left(\mathbb{G}_{t+1}^k\left(\boldsymbol{\mu}_t\right) = G \mid A_2 = \boldsymbol{a}_2\right)$$

$$\approx \sum_{\boldsymbol{a}_3 \in \mathcal{A}^k(G,G',c)} \prod_{j=1}^d \prod_{m \in [k^*] \cup \{\infty\}} \binom{\sum_i a_{ijm}}{a_{1jm}, \ldots, a_{djm}} \mathbf{1}_{\{\sum_i a_{ijm} = a_{jm}\}}$$

$$\cdot \prod_{i=1}^d \left(P_{\boldsymbol{\pi},\boldsymbol{\mu}}\left(x_{t+1}^1 = x_i \mid x_t^1 = x_j, \deg(v_1) = m\right)\right)^{a_{ijm}}$$

$$\approx \sum_{\boldsymbol{a}_3 \in \mathcal{A}^k(G,G',c)} \prod_{j=1}^d \prod_{m \in [k^*] \cup \{\infty\}} \binom{\sum_i a_{ijm}}{a_{1jm}, \ldots, a_{djm}} \mathbf{1}_{\{\sum_i a_{ijm} = a_{jm}\}}$$

$$\cdot \prod_{i=1}^d \left(\sum_{G'' \in \boldsymbol{\mathcal{G}}^m} P_{\boldsymbol{\pi}}\left(\mathbb{G}_t^m\left(\boldsymbol{\mu}_t\right) = G'' \mid x_t'' = s_j\right) \sum_{u \in \mathcal{U}} \pi_t^m\left(u \mid s_j\right) \cdot P\left(s_i \mid s_j, u, G''\right)\right)^{a_{ijm}}$$

where $\mathbf{1}_{\{\ldots\}}$ denotes the indicator function and where we neglect the potential dependencies between the neighbors of the initial node in the second line. Finally, we arrive at

$$P_{\boldsymbol{\pi},\boldsymbol{\mu}}\left(\mathbb{G}_{t+1}^k\left(\boldsymbol{\mu}_t\right) = G \mid \deg_1 = c_1, \ldots, \deg_{k^*} = c_{k^*}, \deg_\infty = c_\infty, \mathbb{G}_t^k\left(\boldsymbol{\mu}_t\right) = G'\right)$$

$$= \sum_{\boldsymbol{a}_2 \in \mathcal{A}_2^k(G',c)} P_{\boldsymbol{\pi},\boldsymbol{\mu}}\left(A_2 = \boldsymbol{a}_2 \mid \deg_1 = c_1, \ldots, \deg_{k^*} = c_{k^*}, \deg_\infty = c_\infty, \mathbb{G}_t^k\left(\boldsymbol{\mu}_t\right) = G'\right)$$

$$\cdot P_{\boldsymbol{\pi},\boldsymbol{\mu}}\left(\mathbb{G}_{t+1}^k\left(\boldsymbol{\mu}_t\right) = G \mid A_2 = \boldsymbol{a}_2\right)$$

$$\approx \sum_{\boldsymbol{a}_2 \in \mathcal{A}_2^k(G',c)} \frac{\prod_{j=1}^d \prod_{m \in [k^*] \cup \{\infty\}} \left(P\left(\deg(v_1) = m \mid (v_0, v_1) \in E\right) \mu_t^m(s_j)\right)^{a_{jm}}}{\sum_{\boldsymbol{a}_2' \in \mathcal{A}_2^k(G',c)} \prod_{j=1}^d \prod_{m \in [k^*] \cup \{\infty\}} \left(P\left(\deg(v_1) = m \mid (v_0, v_1) \in E\right) \mu_t^m(s_j)\right)^{a_{jm}'}}$$

$$\cdot P_{\boldsymbol{\pi},\boldsymbol{\mu}}\left(\mathbb{G}_{t+1}^k\left(\boldsymbol{\mu}_t\right) = G \mid A_2 = \boldsymbol{a}_2\right)$$

$$\approx \sum_{\boldsymbol{a}_2 \in \mathcal{A}_2^k(G',c)} \frac{\prod_{j=1}^d \prod_{m \in [k^*] \cup \{\infty\}} \left(P\left(\deg(v_1) = m \mid (v_0, v_1) \in E\right) \mu_t^m(s_j)\right)^{a_{jm}}}{\sum_{\boldsymbol{a}_2' \in \mathcal{A}_2^k(G',c)} \prod_{j=1}^d \prod_{m \in [k^*] \cup \{\infty\}} \left(P\left(\deg(v_1) = m \mid (v_0, v_1) \in E\right) \mu_t^m(s_j)\right)^{a_{jm}'}}$$

$$\sum_{\boldsymbol{a}_3 \in \mathcal{A}_3^k(G,G',c)} \prod_{j=1}^d \prod_{m \in [k^*] \cup \{\infty\}} \binom{\sum_i a_{ijm}}{a_{1jm}, \ldots, a_{djm}} \mathbf{1}_{\{\sum_i a_{ijm} = a_{jm}\}}$$

$$\cdot \prod_{i=1}^d \left(\sum_{G'' \in \boldsymbol{\mathcal{G}}^m} P_{\boldsymbol{\pi}}\left(\mathbb{G}_t^m\left(\boldsymbol{\mu}_t\right) = G'' \mid x_t'' = s_j\right) \sum_{u \in \mathcal{U}} \pi_t^m\left(u \mid s_j\right) \cdot P\left(s_i \mid s_j, u, G''\right)\right)^{a_{ijm}}$$

and for the running example of power law graphs we especially obtain

$$P_{\boldsymbol{\pi},\boldsymbol{\mu}}\left(\mathbb{G}_{t+1}^k\left(\boldsymbol{\mu}_t\right) = G \mid \deg_1 = c_1, \ldots, \deg_{k^*} = c_{k^*}, \deg_\infty = c_\infty, \mathbb{G}_t^k\left(\boldsymbol{\mu}_t\right) = G'\right)$$

$$\approx \sum_{\boldsymbol{a}_2 \in \boldsymbol{\mathcal{A}}_2^k(G',c)} \frac{\prod_{j=1}^d \left(\mu_t^\infty(s_j)\left(1 - \sum_{m'=1}^{k^*} \frac{(m')^{1-\gamma}}{\zeta(\gamma-1)}\right)\right)^{a_{j\infty}} \prod_{m=1}^{k^*} \left(\frac{m^{1-\gamma}\mu_t^m(s_j)}{\zeta(\gamma-1)}\right)^{a_{jm}}}{\sum_{\boldsymbol{a}_2' \in \boldsymbol{\mathcal{A}}_2^k(G',c)} \prod_j \left(\mu_t^\infty(s_j)\left(1 - \sum_{m'=1}^{k^*}\frac{(m')^{1-\gamma}}{\zeta(\gamma-1)}\right)\right)^{a_{j\infty}'} \prod_{m=1}^{k^*}\left(\frac{m^{1-\gamma}\mu_t^m(s_j)}{\zeta(\gamma-1)}\right)^{a_{jm}'}}$$

$$\sum_{\boldsymbol{a}_3 \in \boldsymbol{\mathcal{A}}_3^k(G,G',c)} \prod_{j=1}^d \prod_{m\in[k^*]\cup\{\infty\}} \binom{\sum_i a_{ijm}}{a_{1jm},\ldots,a_{djm}} \mathbf{1}_{\{\sum_i a_{ijm}=a_{jm}\}}$$

$$\cdot \prod_{i=1}^d \left(\sum_{G''\in\boldsymbol{\mathcal{G}}^m} P_{\boldsymbol{\pi}}\left(\mathbb{G}_t^m(\boldsymbol{\mu}_t)=G'' \mid x_t''=s_j\right)\sum_{u\in\mathcal{U}}\pi_t^m(u\mid s_j)\cdot P\left(s_i\mid s_j,u,G''\right)\right)^{a_{ijm}}$$

$$= \sum_{\boldsymbol{a}_3 \in \boldsymbol{\mathcal{A}}_3^k(G,G',c)} \frac{\prod_{j=1}^d \left(\mu_t^\infty(s_j)\left(1 - \sum_{m'=1}^{k^*} \frac{(m')^{1-\gamma}}{\zeta(\gamma-1)}\right)\right)^{a_{j\infty}} \prod_{m=1}^{k^*} \left(\frac{m^{1-\gamma}\mu_t^m(s_j)}{\zeta(\gamma-1)}\right)^{a_{jm}}}{\sum_{\boldsymbol{a}_2' \in \boldsymbol{\mathcal{A}}_2^k(G',c)} \prod_j \left(\mu_t^\infty(s_j)\left(1 - \sum_{m'=1}^{k^*}\frac{(m')^{1-\gamma}}{\zeta(\gamma-1)}\right)\right)^{a_{j\infty}'} \prod_{m=1}^{k^*}\left(\frac{m^{1-\gamma}\mu_t^m(s_j)}{\zeta(\gamma-1)}\right)^{a_{jm}'}}$$

$$\prod_{j=1}^d \prod_{m\in[k^*]\cup\{\infty\}} \binom{\sum_i a_{ijm}}{a_{1jm},\ldots,a_{djm}}$$

$$\cdot \prod_{i=1}^d \left(\sum_{G''\in\boldsymbol{\mathcal{G}}^m} P_{\boldsymbol{\pi}}\left(\mathbb{G}_t^m(\boldsymbol{\mu}_t)=G'' \mid x_t''=s_j\right)\sum_{u\in\mathcal{U}}\pi_t^m(u\mid s_j)\cdot P\left(s_i\mid s_j,u,G''\right)\right)^{a_{ijm}} .$$

**Resulting Approximation**   Eventually, we obtain the approximation

$$P_{\boldsymbol{\pi},\boldsymbol{\mu}}\left(\mathbb{G}_{t+1}^k(\boldsymbol{\mu}_t)=G, x_{t+1}=x\right)$$

$$\approx \sum_{G'\in\boldsymbol{\mathcal{G}}^k} \sum_{x'\in\mathcal{X}} P_{\boldsymbol{\pi},\boldsymbol{\mu}}\left(\mathbb{G}_t^k(\boldsymbol{\mu}_t)=G', x_t=x'\right)\left[\sum_{u\in\mathcal{U}}\pi^k(u\mid x')P(x\mid x',u,G')\right]$$

$$\cdot \sum_{c\in\boldsymbol{\mathcal{C}}^k}\left[\sum_{\boldsymbol{a}_2\in\boldsymbol{\mathcal{A}}^k(G',c)}\prod_{j=1}^d \binom{g_j'}{a_{j1},\ldots,a_{j\infty}}\prod_{m\in[k^*]\cup\{\infty\}}\right.$$

$$\cdot \left(\frac{P\left(\deg(v_1)=m \mid (v_0,v_1)\in E\right)\mu_t^m(s_j)}{P\left(\deg(v_1)>k^* \mid (v_0,v_1)\in E\right)\mu_t^\infty(s_j)+\sum_{k=1}^{k^*}P\left(\deg(v_1)=k \mid (v_0,v_1)\in E\right)\mu_t^k(s_j)}\right)^{a_{jm}}\right]$$

$$\cdot \frac{1}{\sum_{\boldsymbol{a}_2'\in\boldsymbol{\mathcal{A}}_2^k(G',c)}\prod_{j,m}\left(P\left(\deg(v_1)=m \mid (v_0,v_1)\in E\right)\mu_t^m(s_j)\right)^{a_{jm}'}}$$

$$\sum_{\boldsymbol{a}_3\in\boldsymbol{\mathcal{A}}_3^k(G,G',c)}\prod_{j=1}^d\prod_{m\in[k^*]\cup\{\infty\}}\binom{\sum_i a_{ijm}}{a_{1jm},\ldots,a_{djm}}\prod_{i=1}^d\left(P\left(\deg(v_1)=m \mid (v_0,v_1)\in E\right)\mu_t^m(s_j)\right)^{a_{ijm}}$$

$$\cdot \left(\sum_{G''\in\boldsymbol{\mathcal{G}}^m}P_{\boldsymbol{\pi}}\left(\mathbb{G}_t^m(\boldsymbol{\mu}_t)=G'' \mid x_t''=s_j\right)\sum_{u\in\mathcal{U}}\pi_t^m(u\mid s_j)\cdot P\left(s_i\mid s_j,u,G''\right)\right)^{a_{ijm}}$$

which, for the power law running example, can be reformulated as

$$P_{\boldsymbol{\pi},\boldsymbol{\mu}}\left(\mathbb{G}_{t+1}^k(\boldsymbol{\mu}_t)=G, x_{t+1}=x\right)$$

$$\approx \sum_{G'\in\boldsymbol{\mathcal{G}}^k} \sum_{x'\in\mathcal{X}} P_{\boldsymbol{\pi},\boldsymbol{\mu}}\left(\mathbb{G}_t^k(\boldsymbol{\mu}_t)=G', x_t=x'\right)\left[\sum_{u\in\mathcal{U}}\pi^k(u\mid x')P(x\mid x',u,G')\right]$$

$$\cdot \sum_{c\in\boldsymbol{\mathcal{C}}^k}\left[\sum_{\boldsymbol{a}_2\in\boldsymbol{\mathcal{A}}^k(G',c)}\prod_{j=1}^d \binom{g_j'}{a_{j1},\ldots,a_{j\infty}}\right.$$

$$\left.\cdot \prod_{m\in[k^*]\cup\{\infty\}}\left(\frac{m^{1-\gamma}\mu_t^m(s_j)}{\left[\sum_{\ell=k^*+1}^\infty \ell^{1-\gamma}\right]\mu_t^\infty(s_j)+\sum_{h=1}^{k^*}h^{1-\gamma}\mu_t^h(s_j)}\right)^{a_{jm}}\right]$$

$$\cdot \sum_{\boldsymbol{a}_3 \in \mathcal{A}_3^k(G,G',c)} \frac{\prod_{j=1}^d \left( \mu_t^\infty(s_j) \left( 1 - \sum_{m'=1}^{k^*} \frac{(m')^{1-\gamma}}{\zeta(\gamma-1)} \right) \right)^{a_{j\infty}} \prod_{m=1}^{k^*} \left( \frac{m^{1-\gamma} \mu_t^m(s_j)}{\zeta(\gamma-1)} \right)^{a_{jm}}}{\sum_{\boldsymbol{a}_2' \in \mathcal{A}_2^k(G',c)} \prod_j \left( \mu_t^\infty(s_j) \left( 1 - \sum_{m'=1}^{k^*} \frac{(m')^{1-\gamma}}{\zeta(\gamma-1)} \right) \right)^{a'_{j\infty}} \prod_{m=1}^{k^*} \left( \frac{m^{1-\gamma} \mu_t^m(s_j)}{\zeta(\gamma-1)} \right)^{a'_{jm}}}$$

$$\prod_{j=1}^d \prod_{m \in [k^*] \cup \{\infty\}} \binom{\sum_i a_{ijm}}{a_{1jm}, \ldots, a_{djm}}$$

$$\cdot \prod_{i=1}^d \left( \sum_{G'' \in \boldsymbol{\mathcal{G}}^m} P_{\boldsymbol{\pi}} \left( \mathbb{G}_t^m(\boldsymbol{\mu}_t) = G'' \mid x_t'' = s_j \right) \sum_{u \in \mathcal{U}} \pi_t^m(u \mid s_j) \cdot P(s_i \mid s_j, u, G'') \right)^{a_{ijm}}$$

For notational convenience, define for each $j \in [d]$ and $m \in [k^*] \cup \{\infty\}$

$$p_{jm} := \frac{P(\deg(v_1) = m \mid (v_0, v_1) \in E) \, \mu_t^m(s_j)}{P(\deg(v_1) > k^* \mid (v_0, v_1) \in E) \, \mu_t^\infty(s_j) + \sum_{k=1}^{k^*} P(\deg(v_1) = k \mid (v_0, v_1) \in E) \, \mu_t^k(s_j)}$$

and for each $i, j \in [d]$ and $m \in [k^*] \cup \{\infty\}$

$$p_{ijm} := P(\deg(v_1) = m \mid (v_0, v_1) \in E) \, \mu_t^m(s_j)$$

$$\cdot \sum_{G'' \in \boldsymbol{\mathcal{G}}^m} P_{\boldsymbol{\pi}} \left( \mathbb{G}_t^m(\boldsymbol{\mu}_t) = G'' \mid x_t'' = s_j \right) \sum_{u \in \mathcal{U}} \pi_t^m(u \mid s_j) \cdot P(s_i \mid s_j, u, G'') \, .$$

Then, the extensive approximation can be rewritten more compactly as

$$P_{\boldsymbol{\pi},\boldsymbol{\mu}} \left( \mathbb{G}_{t+1}^k(\boldsymbol{\mu}_t) = G, x_{t+1} = x \right)$$

$$\approx \sum_{G' \in \boldsymbol{\mathcal{G}}^k} \sum_{x' \in \mathcal{X}} P_{\boldsymbol{\pi},\boldsymbol{\mu}} \left( \mathbb{G}_t^k(\boldsymbol{\mu}_t) = G', x_t = x' \right) \left[ \sum_{u \in \mathcal{U}} \pi^k(u \mid x') P(x \mid x', u, G') \right]$$

$$\cdot \sum_{c \in \boldsymbol{\mathcal{C}}^k} \left[ \sum_{\boldsymbol{a}_2 \in \mathcal{A}_2^k(G',c)} \prod_{j=1}^d \binom{g_j'}{a_{j1}, \ldots, a_{j\infty}} \prod_{m \in [k^*] \cup \{\infty\}} p_{jm}^{a_{jm}} \right]$$

$$\cdot \frac{\sum_{\boldsymbol{a}_3 \in \mathcal{A}_3^k(G,G',c)} \prod_{j=1}^d \prod_{m \in [k^*] \cup \{\infty\}} \binom{\sum_i a_{ijm}}{a_{1jm}, \ldots, a_{djm}} \prod_{i=1}^d p_{ijm}^{a_{ijm}}}{\sum_{\boldsymbol{a}_2 \in \mathcal{A}_2^k(G',c)} \prod_{j,m} \left( P(\deg(v_1) = m \mid (v_0, v_1) \in E) \, \mu_t^m(s_j) \right)^{a_{jm}}} \, .$$

Furthermore, we introduce

$$\boldsymbol{p}_{2,j} := (p_{j1}, \ldots, p_{jk^*}, p_{j\infty}) \quad \text{and} \quad \boldsymbol{a}_{2,j} := (a_{j1}, \ldots, a_{jk^*}, a_{j\infty})$$

for every $j \in [d]$ and similarly we define

$$\boldsymbol{p}_{3,jm} := (p_{1jm}, \ldots, p_{djm}) \quad \text{and} \quad \boldsymbol{a}_{3,jm} := (a_{1jm}, \ldots, a_{djm})$$

for every tuple $(j, m) \in [d] \times ([k^*] \cup \{\infty\})$. Then, the extensive approximation can be formulated as

$$P_{\boldsymbol{\pi},\boldsymbol{\mu}} \left( \mathbb{G}_{t+1}^k(\boldsymbol{\mu}_t) = G, x_{t+1} = x \right)$$

$$\approx \sum_{G' \in \boldsymbol{\mathcal{G}}^k} \sum_{x' \in \mathcal{X}} P_{\boldsymbol{\pi},\boldsymbol{\mu}} \left( \mathbb{G}_t^k(\boldsymbol{\mu}_t) = G', x_t = x' \right) \left[ \sum_{u \in \mathcal{U}} \pi^k(u \mid x') P(x \mid x', u, G') \right]$$

$$\cdot \frac{\sum_{c \in \boldsymbol{\mathcal{C}}^k} \left[ \sum_{\boldsymbol{a}_2 \in \mathcal{A}_2^k(G',c)} \prod_j \text{Mult}_{\boldsymbol{p}_{2,j}}(\boldsymbol{a}_{2,j}) \right] \sum_{\boldsymbol{a}_3 \in \mathcal{A}_3^k(G,G',c)} \prod_{j,m} \text{Mult}_{\boldsymbol{p}_{3,jm}}(\boldsymbol{a}_{3,jm})}{\sum_{\boldsymbol{a}_2 \in \mathcal{A}_2^k(G',c)} \prod_{j,m} \left( P(\deg(v_1) = m \mid (v_0, v_1) \in E) \, \mu_t^m(s_j) \right)^{a_{jm}}} \, .$$

## C  SIMULATION DETAILS

We use MARLlib 1.0.0 (Hu et al., 2023a) building on RLlib 1.8.0 (Apache-2.0 license) (Liang et al., 2018) and its PPO implementation (Schulman et al., 2017) for IPPO and our algorithms. For our experiments, we used around $80\,000$ core hours on Intel Xeon Platinum 9242 CPUs, and each training run usually took a single day of training on up to 96 parallel CPU cores. For the policies we used two hidden layers of 256 nodes with $\tanh$ activations. We used a discount factor of $\gamma = 0.99$ with GAE $\lambda = 1.0$, and training and minibatch sizes of 4000 and 1000, performing 5 updates per training batch. The KL coefficient and clip parameter were set to $0.2$, with a KL target of $0.03$. The learning rate was set to $0.00005$. The problem details are found in the following.

**Susceptible-Infected-Susceptible (SIS).**  In the SIS model with state space $\mathcal{X} \coloneqq \{S, I\}$, agents are either infected ($I$) or susceptible to a virus ($S$). At each time step $t \in \mathcal{T}$, agents either protect themselves ($P$) or not ($\bar{P}$) which is formalized by the action space $\mathcal{U} \coloneqq \{P, \bar{P}\}$. As usual, the game terminates at finite terminal time $T \in \mathbb{N}$ which can be interpreted as the time when a cure for the virus is found. Therefore, it remains to specify the transition dynamics. Susceptible agents who protect themselves at time $t$ also remain susceptible at time $t + 1$, i.e.

$$P^k(S \mid S, P, G) = 1 \quad \text{and} \quad P^k(I \mid S, P, G) = 0,$$

irrespective of their degree $k$ and neighborhood $G$. On the other hand, if a susceptible agent chooses action $\bar{P}$, the transition dynamics are

$$P^k(I \mid S, \bar{P}, G) = \rho_I \cdot G(I) \cdot \left( \frac{2}{1 + \exp(-k/2)} - 1 \right)$$

and $P^k(S \mid S, \bar{P}, G) = 1 - P^k(I \mid S, \bar{P}, G)$, correspondingly, and where $\rho_I > 0$ is a fixed infection rate. Apart from that, infected agents recover with some fixed recovery rate $1 \geq \rho_R \geq 0$, independent of their action and degree, which means that

$$P^k(S \mid I, \bar{P}, G) = P^k(S \mid I, P, G) = \rho_R.$$

To complete the model, the reward per agent taking action $u \in \mathcal{U}$ in state $x \in \mathcal{X}$ at each time $t$ is

$$r(x, u) = -c_P \cdot \mathbf{1}_P(u) - c_I \cdot \mathbf{1}_I(x),$$

where the cooperative objective $J$ is obtained by talking the average reward over all agents and summing up over all time points. Here, $c_P$ and $c_I$ denote the constant costs of protecting oneself and being infected, respectively. In our experiments from the main text, the chosen parameter values are $\mu_0(I) = 0.4, \mu_0(S) = 0.6, T = 50, \rho_I = 0.4, \rho_R = 0.1, c_P = 0.5$, and $c_I = 1$.

**Susceptible-Infected-Recovered (SIR).**  In the SIR model, we extend the state space from the SIS by the recovered state $R$ and obtain $\mathcal{X} \coloneqq \{S, I, R\}$. As only infected agents can recover, the transition dynamics of the SIS model are modified by

$$P^k(R \mid I, \bar{P}, G) = P^k(R \mid I, P, G) = \rho_R$$

and

$$P^k(R \mid R, \bar{P}, G) = P^k(R \mid R, P, G) = 1,$$

to formalize that recovered agents cannot become susceptible or infected again. The rewards and hence objective remain the same as in the SIS model. In the experiments, we set the parameter values $\mu_0(I) = 0.1, \mu_0(S) = 0.9, T = 50, \rho_I = 0.1, \rho_R = 0.02, c_P = 0.25$, and $c_I = 1$.

**Graph coloring (Color).**  In this problem, the state space consists of five colors $\mathcal{X} \coloneqq \{x_1, x_2, x_3, x_4, x_5\}$ allocated on a circle. Agents can move from the current color to the next color on the left ($\ell$), to the next one on the right ($r$), or stay at their current color ($s$) such that the action space is $\mathcal{U} \coloneqq \{\ell, r, s\}$. The group of agents is also supposed to come close to a target distribution $\nu \in \mathcal{P}(\mathcal{X})$. To keep notations manageable, we make the auxiliary definition

$$\tilde{G}_k \coloneqq \min(1, G^2 \cdot \rho_d \cdot \exp(-2/k)),$$

where $\rho_d > 0$ is a constant noise factor. The following three matrices specify the transition dynamics, where the row is the current agent color and the column is the next agent color:

$$P^k(\cdot \mid \cdot, \ell, G) = \begin{pmatrix} \tilde{G}_k(x_1)/2 & 0 & 0 & \tilde{G}_k(x_1)/2 & 1 - \tilde{G}_k(x_1) \\ 1 - \tilde{G}_k(x_2) & \tilde{G}_k(x_2)/2 & 0 & 0 & \tilde{G}_k(x_2)/2 \\ \tilde{G}_k(x_3)/2 & 1 - \tilde{G}_k(x_3) & \tilde{G}_k(x_3)/2 & 0 & 0 \\ 0 & \tilde{G}_k(x_4)/2 & 1 - \tilde{G}_k(x_4) & \tilde{G}_k(x_4)/2 & 0 \\ 0 & 0 & \tilde{G}_k(x_5)/2 & 1 - \tilde{G}_k(x_5) & \tilde{G}_k(x_5)/2 \end{pmatrix}$$

and

$$P^k(\cdot \mid \cdot, s, G) = \begin{pmatrix} 1 - \tilde{G}_k(x_1) & \tilde{G}_k(x_1)/2 & 0 & 0 & \tilde{G}_k(x_1)/2 \\ \tilde{G}_k(x_2)/2 & 1 - \tilde{G}_k(x_2) & \tilde{G}_k(x_2)/2 & 0 & 0 \\ 0 & \tilde{G}_k(x_3)/2 & 1 - \tilde{G}_k(x_3) & \tilde{G}_k(x_3)/2 & 0 \\ 0 & 0 & \tilde{G}_k(x_4)/2 & 1 - \tilde{G}_k(x_4) & \tilde{G}_k(x_4)/2 \\ \tilde{G}_k(x_5)/2 & 0 & 0 & \tilde{G}_k(x_5)/2 & 1 - \tilde{G}_k(x_5) \end{pmatrix}$$

and

$$P^k(\cdot \mid \cdot, r, G) = \begin{pmatrix} \tilde{G}_k(x_1)/2 & 1-\tilde{G}_k(x_1) & \tilde{G}_k(x_1)/2 & 0 & 0 \\ 0 & \tilde{G}_k(x_2)/2 & 1-\tilde{G}_k(x_2) & \tilde{G}_k(x_2)/2 & 0 \\ 0 & 0 & \tilde{G}_k(x_3)/2 & 1-\tilde{G}_k(x_3) & \tilde{G}_k(x_3)/2 \\ \tilde{G}_k(x_4)/2 & 0 & 0 & \tilde{G}_k(x_4)/2 & 1-\tilde{G}_k(x_4) \\ 1-\tilde{G}_k(x_5) & \tilde{G}_k(x_5)/2 & 0 & 0 & \tilde{G}_k(x_5)/2 \end{pmatrix}.$$

The reward in our graph coloring model is defined as

$$r(x_j, u, G) := -\left(\mathbf{1}_\ell(u) + \mathbf{1}_r(u)\right) \cdot c_m - \left(G(x_{j-1}) + G(x_{j+1})\right) \cdot c_d - \sum_{i=1}^5 |\mu(x_i) - \nu(x_i)| \cdot c_\nu,$$

where $c_m, c_d, c_\nu > 0$ are the costs of moving, having neighbors with neighboring colors, and deviating from the target distribution $\nu$, respectively. In our experiments, we choose the parameters $\mu_0 = (1,0,0,0,0), \nu = (0.1, 0.2, 0.4, 0.2, 0.1), T = 20, \rho_d = 0.9, c_m = 0.1, c_d = 0.5,$ and $c_\nu = 1$.

**Rumor.** The state space $\mathcal{X} := \{I, A\}$ in the rumor model consists of the state $A$ where an agent is aware of a rumor and state $I$ where the agent does not know the rumor and is therefore ignorant of the rumor. Agents either spread the rumor $S$ or decide not to do so $\bar{S}$ which results in the action space $\mathcal{U} := \{S, \bar{S}\}$. Since the rumor spreading probability increases with the number of aware numbers who decide to spread the rumor, we work with the extended state space $\mathcal{X}' := \mathcal{X} \cup (\mathcal{X} \times \mathcal{U})$. Then, the transition dynamics are

$$P^k((A,u) \mid A, u, G) = P^k(A \mid (A,u), u, G) = 1, \quad \forall u \in \mathcal{U}, G \in \mathcal{G}^k, k \in \mathbb{N}$$

meaning that aware agents remain aware, and furthermore

$$P^k((I,u) \mid I, u, G) = 1, \quad \forall u \in \mathcal{U}, G \in \mathcal{G}^k, k \in \mathbb{N}$$

and

$$P^k(A \mid (I,u), u, G) = \min\left(1, \rho_A \cdot G((A,S)) \cdot \left(\frac{2}{1+\exp(-k/2)} - 1\right)\right)$$
$$P^k(I \mid (I,u), u, G) = 1 - P^k(A \mid (I,u), u, G).$$

To complete the rumor model, the reward is given by

$$r(x, u, G) = \mathbf{1}_{(A,S)}(x) \cdot \left(r_S \cdot G((I,S)) + r_S \cdot G((I,\bar{S})) - c_S \cdot G((A,S)) - c_S \cdot G((A,\bar{S}))\right)$$

for each agent, where we obtain the overall objective by averaging over the individual rewards. In our experiments, the parameters are chosen as $\mu_0(A) = 0.1, \mu_0(I) = 0.9, T = 50, \rho_A = 0.3, c_S = 16,$ and $r_S = 4$.

