# OpenReview forum: "Learning Cooperative Mean Field Games on Sparse Chung-Lu Graphs"
_ICLR.cc/2025/Conference — Submitted to ICLR 2025_

### Official Review · Reviewer_2rVF · 2024-11-03

**Soundness:** 2
**Presentation:** 2
**Contribution:** 2
**Rating:** 5
**Confidence:** 4

**Summary:**

This paper proposes a learning framework for cooperative behaviors among large agent populations on sparse graphs. Authors propose learning algorithms and conduct numerical experiments to demonstrate the performance.

**Strengths:**

Overall, this paper is well written and structured. The numerical results are thorough.

**Weaknesses:**

1. The research gap is not quite clear. The problem formulation of  cooperative mean field games is similar to mean field control (MFC). Authors should clearly present the relationship between the proposed work, MFC and multi-agent cooperative learning or compare the difference corresponding to the problem formulation or set-up and summarize them in a table. I feel like the proposed model in this paper is essentially an MFC over sparse graphs.

2. Is the solution concept an extension of equilibria in multi-agent cooperative games when N goes to infinity?

3. Sparse graphs have been studied in many existing works (Leaning Mean Field Games on Sparse Graphs: A Hybrid Graphex Approach, 2024). Authors should highlight the contribution and novelty of the proposed sparse graphs in this work and demonstrate their superiority over other frameworks in numerical results, e.g., the comparison between Chung-Lu graphs proposed in this work and those in Leaning Mean Field Games on Sparse Graphs: A Hybrid Graphex Approach, 2024.

4. The solution properties of the cooperative mean field game (existence and uniqueness) are not discussed in this work.

5. The proposed two algorithms are very similar to some classic multi-agent reinforcement learning algorithms (e.g., MF-MARL, 2018). Authors should compare the difference between these algorithms in a table (e.g., complexity, setup, etc) to highlight their contribution. I feel like MF-MARL, 2018 also solves a multi-agent cooperative game.

6. The theoretical analysis regarding algorithm performance (convergence, time complexity) is not discussed.

**Questions:**

Please see the list of questions in weaknesses.

---

> ### Author Response · Authors · 2024-11-20
> **Response to Reviewer 2rVF**
>
> We thank the reviewer for the constructive criticism on our work. The points raised by the reviewer are answered in the order they were brought up in.
>
> ---
>
> *W1: The research gap is not quite clear. The problem formulation of cooperative mean field games is similar to mean field control (MFC). Authors should clearly present the relationship between the proposed work, MFC and multi-agent cooperative learning or compare the difference corresponding to the problem formulation or set-up and summarize them in a table. I feel like the proposed model in this paper is essentially an MFC over sparse graphs.*
>
> A: We use the term “cooperative MFGs” as an equivalent term for MFC. The difference of our work to general multi-agent cooperative learning is that we exploit MF techniques to obtain scalable learning solutions, even for systems with millions or billions of agents. In contrast to standard MFC formulations, our model addresses challenging large, sparse agent networks by proposing a two systems approximation. Our work can thus be seen as an extension of standard MFC to the case of these difficult large agent graphs.
>
> ---
>
> *W2: Is the solution concept an extension of equilibria in multi-agent cooperative games when N goes to infinity?*
>
> A: Yes, our solution concept is to some extent a limiting version of multi-agent cooperative games on networks. Providing a scalable learning approach through the limiting model is a key motivation for our paper.
>
> ---
>
> *W3: Sparse graphs have been studied in many existing works (Leaning Mean Field Games on Sparse Graphs: A Hybrid Graphex Approach, 2024). Authors should highlight the contribution and novelty of the proposed sparse graphs in this work and demonstrate their superiority over other frameworks in numerical results, e.g., the comparison between Chung-Lu graphs proposed in this work and those in Leaning Mean Field Games on Sparse Graphs: A Hybrid Graphex Approach, 2024.*
>
> A: The graphs studied in (Fabian et al., 2024) are at most moderately sparse where the expected average degree diverges to infinity. The graphs in our work are sparser and CLCMFGs can depict these graph structures with finite first, but infinite second moment. We demonstrate the superiority over state-of-the-art frameworks such as LPGMFGs (Fabian et al., 2023) and GXMFGs (Fabian et al., 2024) on four example problems for eight different real-world networks, see Tables 1 and 2 for empirical results. These comprehensive empirical evaluations depict the significant advantages of CLCMFGs over state-of-the-art methods like GXMFGs.
>
> ---
>
> *W4: The solution properties of the cooperative mean field game (existence and uniqueness) are not discussed in this work.*
>
> A: Deriving solution properties will likely require more restrictive assumptions on the model dynamics. The focus of our work is the practical two systems approximation and designing applicable learning algorithms which we validate by a comprehensive empirical evaluation.
>
> ---
>
> *W5: The proposed two algorithms are very similar to some classic multi-agent reinforcement learning algorithms (e.g., MF-MARL, 2018). Authors should compare the difference between these algorithms in a table (e.g., complexity, setup, etc) to highlight their contribution. I feel like MF-MARL, 2018 also solves a multi-agent cooperative game.*
>
> A: In contrast to the many existing approaches for solving multi-agent cooperative games, our two algorithms are specifically designed to learn approximately optimal agent behavior in large, sparse, and realistic network. Thus, we aim to solve this relevant and challenging subclass of problems with tailored methods instead of addressing general multi-agent cooperative games. As our empirical results show, these tailored CLCMFG methods yield significantly better empirical results than existing approaches.
>
> ---
>
> *W6: The theoretical analysis regarding algorithm performance (convergence, time complexity) is not discussed.*
>
> A: Due to the scope of our paper, we focus on an extensive empirical evaluation of our learning algorithms and existing approaches and leave a theoretical algorithm analysis to future work.

---

> > ### Comment · Reviewer_2rVF · 2024-11-20
> >
> > Thanks for authors' responses. I appreciate authors' clarification regarding MFC (W1), solution concepts (W2) and the sparse graphs (W3) in previous work. However, my concern regarding solution properties (W4) is not well addressed. Solution properties corresponding to MFC problems are an important topic. Also, I am not convinced by the clarification on the difference between the proposed algorithms and existing mean field algorithms for multi-agent cooperative games (W5, W6). Therefore, I will keep my original score.

---

> > > ### Author Response · Authors · 2024-11-26
> > >
> > > We appreciate the response and thank the reviewer for considering our answers. In the following, we provide further explanations and hope that this addresses the remaining concerns of the reviewer.
> > >
> > > ---
> > >
> > > **Differences to existing MF algorithms for multi-agent cooperative games:** The crucial difference to existing MF models and algorithms is that the agents in our model are connected by a complicated, realistic graph structure. This network topology renders existing MF approaches useless, as standard MF methods usually assume that there is one representative agent for the entirety of agents. In our setup with realistic networks, however, the degree distributions are highly heterogeneous which implies that we have to consider multiple representative agents for different low degrees and for those with (almost) infinitely many neighbors. To achieve this, we propose the two systems approximation which is the foundation for our learning algorithms. Another challenge is that the mean field neighborhoods of the low degree agents remain stochastic in the limit. In contrast, standard MF learning algorithms assume that the mean field becomes deterministic in the limit which makes them unsuitable in realistic network scenarios. Our learning approach overcomes this limitation of existing work by leveraging the two systems approximation to obtain practically useful learning techniques, as Tables 1 and 2 demonstrate.
> > >
> > > We point to our previous answer on W3 for the differences of CLCMFGs to LPGMFGs and GXMFGs. If further clarifications are requested, we would be happy to provide them.
> > >
> > > ---
> > >
> > > **Solution properties (W4):** We do conduct a theoretical analysis which proves that the mean field and objective in the finite system converge to the ones in limiting system (Theorem 1, Proposition 1) and that an optimal policy in the limiting system is also optimal in all finite systems of sufficient size (Corollary 1). We furthermore highlight the mathematical and computational challenges of the realistic networks considered in our work (Lemma 1) and derive a detailed extensive approximation to evaluate the accuracy of our two systems approximation empirically.
> > >
> > > On the other hand, we point to the existing MF learning literature which often proposes empirically successful algorithms without extensive theoretical results. Notable and well-published examples include the papers [A], [B] and [C].
> > >
> > > ---
> > >
> > > References:
> > >
> > > [A] Ruthotto, Lars, et al. "A machine learning framework for solving high-dimensional mean field game and mean field control problems." *Proceedings of the National Academy of Sciences*, 2020.
> > >
> > > [B] Perrin, Sarah, et al. "Mean Field Games Flock! The Reinforcement Learning Way." *IJCAI*, 2021.
> > >
> > > [C] Laurière, Mathieu, et al. "Scalable deep reinforcement learning algorithms for mean field games." *ICML*, 2022.

---

### Official Review · Reviewer_3nHQ · 2024-11-04

**Soundness:** 2
**Presentation:** 2
**Contribution:** 3
**Rating:** 5
**Confidence:** 3

**Summary:**

This paper aims to model large agents on networks and proposes a cooperative mean field game over sparse graphs like Chung-Lu (CL). Two systems approximation is developed to describe how agents with different degrees of connections are connected. Two learning algorithms are proposed, namely, MFC MDP and single-agent RL.

**Strengths:**

The paper makes a unique contribution to large agents interacting over sparse graphs, given that the majority of existing literature on asymmetric interactions of mean field games (MFG) is focused on dense graphs.

**Weaknesses:**

Motivation: In practice, how could one tell if graphon MFGs should be used or the proposed game over CL graphs should be used?

Model contribution: It remains unclear whether this paper makes more contributions to MFGs or graph theory, as the paper spends more space on introducing CL graphs and very little space on cooperative MFG.

Model formulation: It seems this paper proposed a mean field control (MFC) problem over sparse graphs. But the formulation of the proposed model seems different from classical MFC. The finite model is introduced, but the formulation of the limiting system is unclear, partly because the formulation of the limiting system is not presented in detail. What is the exact formulation of the proposed limiting model? How is it related to MFC? It seems the proposed limiting model is simply an MDP with population as the agent. Could the authors clarify the difference between the proposed cooperative MFG here and a generic MFC over sparse graphs?

Learning algorithms: How are these algorithms different from existing scalable learning algorithms for MFGs?
How is the threshold $k^\ast$ determined in practice?
What is the convergence of the learning algorithm?

Examples: The numerical examples can also be applied to graphon MFGs. Could the authors compare the difference of the outputs using graphon MFGs v.s. using the proposed method? What’s the advantage of using CL graphs here?

**Questions:**

How is the CL graph embedded into the cooperative MFGs?

This paper could better motivate why cooperative MFGs on sparse graphs are important in real-world and computation. For example, why is the embedding of a sparse graph challenging? How is it reflected in the formulation, algorithm design, and convergence?

P4 L170: The statement “All agents with degree k share a common policy at all time points.” Is this an assumption or a fact? If latter, could the authors clarify why that's the case?

P4 L189: “Our model also covers reward functions with actions as inputs by using an extended state space and splitting each step into two.” Could the authors detail how this is done for a regular state-action dependent reward?

P4 L203 “Large CL graphs under Assumption 1 have a locally tree-like structure.” Could the authors clarify how the tree-like structure would help with theoretical results?

---

> ### Author Response · Authors · 2024-11-20
> **Response to Reviewer 3nHQ (1/2)**
>
> We thank the reviewer for the detailed review and the constructive feedback. The points raised by the reviewer are answered in the order they were brought up in.
>
> ---
>
> *W1: Motivation: In practice, how could one tell if graphon MFGs should be used or the proposed game over CL graphs should be used?*
>
> A: The graphon and Lp graphon MFG frameworks are unrealistic for most real-world networks, since each agent is assumed to have infinitely many connections in the limit. The choice between graphex MFG and CLMFGs depends on the network under consideration. As a rule of thumb, the GXMFG model is better for networks with a thick tail, e.g. power laws between one and two, while the CLMFG is more suitable for degree distributions with a thinner tail such as power laws between two and three. Here, a log-log plot of the degree distribution (as in Figure 1) gives a first intuition.
>
> ---
>
> *W2: Model contribution: It remains unclear whether this paper makes more contributions to MFGs or graph theory, as the paper spends more space on introducing CL graphs and very little space on cooperative MFG.*
>
> A: Our contributions are to MFGs and their corresponding learning algorithms. While we do not contribute to graph theory, it is important to understand the characteristics and challenges of modelling realistic networks. These existing graph theoretical concepts and insights are the foundation for our MFG contributions and learning algorithms. Consequently, we thoroughly introduce concepts from graph theory such as CL graphs to display our train of thought.
>
> ---
>
> *W3: Model formulation: It seems this paper proposed a mean field control (MFC) problem over sparse graphs. But the formulation of the proposed model seems different from classical MFC. The finite model is introduced, but the formulation of the limiting system is unclear, partly because the formulation of the limiting system is not presented in detail. What is the exact formulation of the proposed limiting model? How is it related to MFC? It seems the proposed limiting model is simply an MDP with population as the agent. Could the authors clarify the difference between the proposed cooperative MFG here and a generic MFC over sparse graphs?*
>
> A: We use the term “cooperative MFG” as a synonym for MFC. First, we introduce the exact limiting system which is computationally intractable for realistic sparse networks and CL graphs, as we show in Lemma 1. The main reason is that the system consists of both low and high degree agents, and that the MF neighborhood for low degree agents is not deterministic (as in standard MFC) but remains stochastic in the limit.  Our novel two systems approximation overcomes the computational intractability of the exact limiting model by a two systems formulation specifically targeting the challenging case of highly heterogeneous agent degrees.
>
> ---
>
> *W4: Learning algorithms: How are these algorithms different from existing scalable learning algorithms for MFGs? How is the threshold* $k^*$ *determined in practice? What is the convergence of the learning algorithm?*
>
> A: The crucial difference to existing work, e.g. LPGMFGs or GXMFGs, is that we consider challenging sparse realistic graphs which require a tailored learning approach based on the two systems approximation. As our comprehensive empirical evaluations show (Tables 1 and 2), this new learning approach yields a substantially higher accuracy than existing state-of-the-art MFG learning methods.  We use the threshold $k^*=10$, except for GXMFGS and CLCMFG* we set $k^*=4$ due to their high computational complexity, for an initial analysis which significantly outperforms existing methods. It seems promising to further investigate the tradeoff in $k^*$ between a higher approximation accuracy for high $k^*$ and the increased action space dimension of the resulting MDP.
>
> ---
>
> *W5: Examples: The numerical examples can also be applied to graphon MFGs. Could the authors compare the difference of the outputs using graphon MFGs v.s. using the proposed method? What’s the advantage of using CL graphs here?*
>
> A: Graphon MFGs are a subclass of the more general Lp graphon MFGs. Since the empirical performance of LPFGMFGs is not convincing, we exclude their GMFG subclass. The main advantage of using CL graphs and the resulting CLMFG model are that they more properly depict many real-world networks. Our empirical evaluation shows how the more accurate model also results in significantly better results compared to the current LPGMFG and GXMFG approaches.
>
> ---

---

> ### Author Response · Authors · 2024-11-20
> **Response to Reviewer 3nHQ (2/2)**
>
> *Q1: How is the CL graph embedded into the cooperative MFGs?*
>
> A: The CL graph is embedded into the cooperative MFG via the two systems approximation. The exact limiting system is computationally not manageable, as Lemma 1 shows. To solve this shortcoming, we design the two systems approximation which stems from the CL graph structure and is based on the observed degree distribution. The two systems approximation accounts for both the many low degree agents and the few but important agents with high degrees.
>
> ---
>
> *Q2: This paper could better motivate why cooperative MFGs on sparse graphs are important in real-world and computation. For example, why is the embedding of a sparse graph challenging? How is it reflected in the formulation, algorithm design, and convergence?*
>
> A: Cooperative MFGs on sparse graphs are important because many applications focus on large, sparse networks with millions or billions of agents such as the human brain or social networks. General MARL methods, however, are often hardly scalable. To circumvent that, we build on the MFG framework and incorporate sparse graphs with finite first and infinite second moment. The challenge is to model both agents with very high degrees as well as agents with only few connections since the MF neighborhoods for low degree agents remain stochastic in the limit in contrast to standard MF models. This is solved by our novel two systems approximation which is key for our algorithm design.
>
> ---
>
> *Q3: P4 L170: The statement “All agents with degree $k$ share a common policy at all time points.” Is this an assumption or a fact? If latter, could the authors clarify why that's the case?*
>
> A: This is a modelling assumption. Since the policies are in general conditioned on the state and neighborhood of the agent, the assumption appears reasonable. For example, the recent work of [A] proves that in the standard MFC framework and two-team generalizations the use of identical policies is almost optimal.
>
> ---
>
> *Q4: P4 L189: “Our model also covers reward functions with actions as inputs by using an extended state space and splitting each step into two.” Could the authors detail how this is done for a regular state-action dependent reward?*
>
> A: As a first step, extend the state space $\mathcal{X}$ to the space $\mathcal{X} \cup (\mathcal{X} \times \mathcal{U})$. Then, each time step is split into two: if an agent at time t is in state x and plays action $u$, than her state at time “$t + 0.5$” is $(u,x)$ and the reward depends on $u$ through the state tuple $(u,x)$. To move to time $t+1$, the agent is then assigned the next state from $\mathcal{X}$ depending on $(u,x)$ and the usual model dynamics.
>
> ---
>
> *Q5: P4 L203 “Large CL graphs under Assumption 1 have a locally tree-like structure.” Could the authors clarify how the tree-like structure would help with theoretical results?*
>
> A: The tree-like structure is fundamental for the proof of Theorem 1 and the subsequent results building on Theorem 1. On an intuitive level, the tree-like structure ensures that the growing graph sequence converges suitably in the local weak sense. Mathematically, the asymptotically tree-like structure of CL graphs is formalized by (Van Der Hofstad, 2024, Theorem 3.18) and allows us to eventually apply (Lacker et al., 2023, Theorem 3.6), see Appendix A.1 for details.
>
> ---
>
> Reference:
>
> [A] Yue Guan et al. "Zero-sum games between mean-field teams: Reachability-based analysis under mean-field sharing." AAAI 2024.

---

### Official Review · Reviewer_SiST · 2024-11-04

**Soundness:** 2
**Presentation:** 3
**Contribution:** 2
**Rating:** 5
**Confidence:** 3

**Summary:**

This paper introduces a novel Chung-Lu Cooperative Mean Field Games (CLCMFG) framework designed for sparse graphs. The model effectively alleviates the computational challenges in applying multi-agent reinforcement learning to large, sparse networks. The authors provide a theoretical foundation for CLCMFGs and introduce a two-systems approximation that simplifies the learning process, reducing computational complexity. The two-systems approximation separates agents into high- and low-degree categories, reducing the computational burden and maintaining accuracy by addressing the heavy-tailed degree distributions typical in sparse graphs. Experiments demonstrate that this approach improves accuracy for sparse network scenarios, overcoming the limitations of prior MFG models, which primarily focus on dense or moderately sparse graphs.

**Strengths:**

1. In contrast to existing MFG models suited to dense graphs, CLCMFGs effectively handle networks with finite average degrees and heavy-tailed distributions, making them particularly relevant for real-world networks like social and communication systems. This advancement broadens the applicability of MFG models to a wider range of network types.
2. Leveraging the two-systems approximation, CLCMFGs avoid the exponential complexity common in MARL for sparse graphs. CLCMFGs address large-degree variance, capturing the impact of high-degree hubs and providing a more accurate representation of network dynamics.
3. The paper conducts a comprehensive empirical evaluation, testing CLCMFGs across multiple problems on both synthetic and real-world network datasets. Results show that CLCMFGs consistently outperform existing LPGMFG and GXMFG methods in accuracy across various settings, including epidemic modeling (SIS/SIR), graph coloring, and rumor spreading.

**Weaknesses:**

1.	While the two-systems approximation is conceptually interesting, there lacks a rigorous error analysis about this approximation, particularly with respect to the choice of $k^*$. Moreover, while the mean field highly depends on the hyperparameter $k^*$, which can significantly affect both the approximation accuracy and computational complexity, the paper has not discussed the choice of $k^*$ in theory or in experiments.
2.	The model is specifically designed around the Chung-Lu graph, which raises questions about whether it can generalize well to other types of sparse networks (if any) and real-world networks. For the latter, it can be observed from Figure1 that there is  still a significant  gap between the proposed model and real-world networks.
3.	The experimental section miss some details, such as a comparison of computational costs for each method in Table 1 and the number of trials for the IPPO method in Figure 3, where results show notable fluctuations. Adding these would clarify computational demands and result reproducibility.

**Questions:**

1.	What key characteristics (e.g., degree distribution, sparsity, high-degree nodes) indicate that a real-world network is suitable for modeling with sparse Chung-Lu graphs?
2.	CLMFC emphasizes computational efficiency through approximation, while CLMFMARL maintains fidelity by interacting with the real graph. Figure 3 shows that CLMFC and CLMFMARL each have advantages in different scenarios. Could you advise on choosing the algorithm for various scenarios?
3.	Can the proposed approach be adapted to other types of random graph generation methods, such as the Barabási-Albert model or the Erdős–Rényi model?
4.	In Figure 1, visualization is used to show that the CL graph closely resembles the real YT network. Would it be possible to incorporate quantitative metrics to compare the performance of different network generation methods?

---

> ### Author Response · Authors · 2024-11-20
> **Response to Reviewer SiST (1/2)**
>
> We thank the reviewer for thoroughly reading our work and the valuable questions. The points raised by the reviewer are answered in the order they were brought up in.
>
> ---
>
> *W1: While the two-systems approximation is conceptually interesting, there lacks a rigorous error analysis about this approximation, particularly with respect to the choice of* $k^*$ . *Moreover, while the mean field highly depends on the hyperparameter* $k^*$, *which can significantly affect both the approximation accuracy and computational complexity, the paper has not discussed the choice of* $k^*$ *in theory or in experiments.*
>
> A: This work focuses on introducing the CLCMFG model and empirically convincing learning algorithms. Our learning methods significantly outperform existing state-of-the-art approaches without hyperparameter tuning of $k^*$ where we choose $k^*=10$ throughout the paper, except for GXMFGS and CLCMFG* with $k^*=4$ due to their high computational complexity. We think that the optimal choice of $k^*$ with respect to accuracy and computational complexity could yield further improvements for our current algorithms, which we leave to future work.
>
> ---
>
> *W2: The model is specifically designed around the Chung-Lu graph, which raises questions about whether it can generalize well to other types of sparse networks (if any) and real-world networks. For the latter, it can be observed from Figure1 that there is still a significant gap between the proposed model and real-world networks.*
>
> A: The model extends to other network types such as Barabasi-Albert or Erdös-Renyi graphs, see the answer to Question 3 below for details. We choose CL graphs as a tool to approximate real-world networks. As the reviewer said, Figure 1 shows that the CL graph and real network are not identical which is not surprising due to the imperfect and highly complex nature of real-world datasets in general. However, Figure 1 also displays that the CL graph is a far better approximation for the real-world network than the Lp graphon or graphex approach. And, as our comprehensive empirical evaluations in Tables 1 and 2 show, CLCMFGs significantly outperform previous state-of-the-art methods on a variety of real-world networks. Consequently, our model and learning algorithm extend well to real-world networks.
>
> ---
>
> *W3: The experimental section misses some details, such as a comparison of computational costs for each method in Table 1 and the number of trials for the IPPO method in Figure 3, where results show notable fluctuations. Adding these would clarify computational demands and result reproducibility.*
>
> A: All experiments shown in Figure 3 correspond to a single trial, and matching with the caption of Table 2, we had training times of approximately 24 hours of training on 96 CPU cores collecting samples in parallel. The comparison of computational cost is similar for IPPO, while for MFC / MFMARL we find faster convergence (as in Table 3, the results are cut-off already after two hours). We believe this comparison however to not be very meaningful, because for MFMARL the environment implementation parallelizes all agents, whereas IPPO in the MARLlib / RLlib implementation uses slow Python dictionaries.
>
> ---

---

> ### Author Response · Authors · 2024-11-20
> **Response to Reviewer SiST (2/2)**
>
> *Q1: What key characteristics (e.g., degree distribution, sparsity, high-degree nodes) indicate that a real-world network is suitable for modeling with sparse Chung-Lu graphs?*
>
> A:  Many real-world networks have a degree distribution which to some extent follows a power law with power law coefficient between two and three. That means that the corresponding graph is sparse and has many nodes with relatively low degrees and a few nodes with very high degrees. This is formalized by the Chung-Lu model, but as we show in the simulations, the algorithms also perform significantly better than current methods on real-world networks.
>
> ---
>
> *Q2: CLMFC emphasizes computational efficiency through approximation, while CLMFMARL maintains fidelity by interacting with the real graph. Figure 3 shows that CLMFC and CLMFMARL each have advantages in different scenarios. Could you advise on choosing the algorithm for various scenarios?*
>
> A: The CLMFC algorithm learns directly on the limiting CLCMFG and therefore scales to graphs of arbitrary size without increasing runtime, but the underlying model has to be known. Since this might not be the case in many applications, we additionally provide CLMFMARL, which is motivated through our theory and interacts with the real graph. Thus, CLMFMARL is a good choice for applications where the model dynamics are not known in advance.
>
> ---
>
> *Q3: Can the proposed approach be adapted to other types of random graph generation methods, such as the Barabasi-Albert model or the Erdös-Renyi model?*
>
> A: Sure, the approach extends to other types of graphs as well. Since the Barabasi-Albert model always generates graphs with power law coefficient three, we decided to focus on the more versatile Chung-Lu graphs capturing any power law above two. For ER graphs, the answer is twofold. If the ER graph is dense, e.g. each node is connected to roughly fifty percent of all other nodes, the case is covered by the existing (LP)GMFG models. However, if the ER graph is ultrasparse, e.g. each node connects on average to four other nodes, our approach can be adapted to cover this case. Then, our two systems approximation essentially reduces to a one system approximation because the system with high degree agents is “empty”. While our algorithms also works in this scenario, we focus on the challenging case of very heterogeneous degree distributions observed in many real-world networks.
>
> ---
>
> *Q4: In Figure 1, visualization is used to show that the CL graph closely resembles the real YT network. Would it be possible to incorporate quantitative metrics to compare the performance of different network generation methods?*
>
> A: We only provide the qualitative network visualizations since the theoretical properties of CL graphs, Lp graphons and graphexes are covered extensively by the existing literature, see e.g. references in the paper. Mathematically, CL graphs can model graphs with finite average expected degree but infinite variance, which both Lp graphons and graphexes are unable to do. The focus of our comprehensive quantitative analysis is on the model dynamics and learning algorithms because these imply which graph model is suitable for approximating the presented real-world networks.

---

### Official Review · Reviewer_TUrs · 2024-11-04

**Soundness:** 3
**Presentation:** 3
**Contribution:** 2
**Rating:** 5
**Confidence:** 4

**Summary:**

This paper aims at learning optimal policies in games with many players in which the players interact through a specific type of sparse graph. After discussing the connection between finite-player situations and the limiting setting when the number of players goes to infinity, two algorithm are described and tested on several examples. Comparison with two previous algorithms is included.

**Strengths:**

The paper is well written. The theory clarifies the link between finite-player games and the limiting game. The experiments cover several examples of large-scale graphs.

**Weaknesses:**

The paper focuses on a special family of graphs. There is no analysis for the learning algorithms. The contributions seem somewhat incremental compared with (Fabian et al., 2023; 2024).

**Questions:**

1. Page 1: “cooperative MFGs” It seems that these are usually called mean field control (MFC) problems. Can you please clarify? (By the way, it seems you use “MFC” on page 6 without explaining its meaning.)

2. Limiting system: Can't this system be reduced to a standard “cooperative MFG” by extending the state? For example, what about considering the “cooperative MFG” in which the state of a representative agent is $(X_t, \mathbb{G}_t)$?

3. Could you explain whether there is any theoretical convergence guarantees for these algorithms?

---

> ### Author Response · Authors · 2024-11-20
> **Response to Reviewer TUrs**
>
> We thank the reviewer for assessing our work and the feedback. The points raised by the reviewer are answered in the order they were brought up in.
>
> ---
>
> *W: The paper focuses on a special family of graphs.*
>
> A: While our initial investigation of the topic starts with Chung-Lu graphs, the algorithms we derive significantly outperform existing methods on various real graphs (Tables 1 and 2) and are thus not limited to Chung-Lu graphs. Furthermore, our two systems approach extends to other families of graphs which we exclude for expositional simplicity.
>
> ---
>
> *W: There is no analysis for the learning algorithms. The contributions seem somewhat incremental compared with (Fabian et al., 2023; 2024)*
>
> A: The graphs considered in this work with finite first moment and infinite second moment are very challenging but also realistic. Theses graphs are harder to incorporate into a mean field framework than the networks considered in (Fabian et al., 2023; 2024), since in (Fabian et al., 2023; 2024) the expected average degree goes to infinity which entails nice theoretical properties. As we demonstrate in our comprehensive empirical analysis in Tables 1 and 2, CLCMFGs yield a substantial performance improvement on various real-world graphs compared to the current state-of-the-art approaches in (Fabian et al., 2023; 2024). We additionally derive a second, novel extensive approximation to analyze the accuracy of our first approximation.
>
> ---
>
> *Q1: Page 1: “cooperative MFGs” It seems that these are usually called mean field control (MFC) problems. Can you please clarify? (By the way, it seems you use “MFC” on page 6 without explaining its meaning.)*
>
> A: The term “MFC” is used interchangeably for “cooperative MFGs” in our paper. We chose the term Chung-Lu cooperative mean field games (CLCMFGs) to emphasize that this work continues the efforts of LPGMFGs (Fabian et al., 2023) and GXMFGs (Fabian et al., 2024) on modelling large agent networks and we wanted to reflect this in the name of CLCMFGs.
>
> ---
> *Q2: Limiting system: Can't this system be reduced to a standard “cooperative MFG” by extending the state? For example, what about considering the “cooperative MFG” in which the state of a representative agent is $(X_t, \mathbb{G}_t)$?*
>
> A: The system cannot be reduced to a standard cooperative MFG/MFC problem. First, one representative agent would not be enough since the finite system contains agents of highly varying degrees. Even if one includes the degree $k$ into the state as well, e.g.  $(X_t, \mathbb{G}_t, k)$, the state evolution is wrongly specified by a standard MF framework. The reason is that the evolution of $\mathbb{G}_t$ highly depends on the neighborhoods of all agents in $\mathbb{G}_t$. Therefore, a state transition kernel would basically need to take the states of all agents in the system individually into account (instead of the aggregated MF), making a proper standard MF formulation infeasible here. Furthermore, for agents with low finite degree, their MF neighborhood does not become deterministic in the limit as in standard MFC but remains stochastic in the limiting model.
>
> ---
>
> *Q3: Could you explain whether there is any theoretical convergence guarantees for these algorithms?*
>
> A: Our evaluation of the CLCMFG based learning algorithms is concerned with the empirical validation of our approach. Theoretical convergence guarantees are outside the scope of this work and are left for future research.

---

> > ### Comment · Reviewer_TUrs · 2024-12-01
> > **Reply**
> >
> > I would like to thank the authors for their response. I will maintain my score.

---

### Author Response · Authors · 2024-11-26
**Response to all Reviewers**

We thank all reviewers for carefully reading our paper and their constructive feedback. We hope that our responses answered all their questions. If something remained unclear, we would be happy to provide further explanations. In case their concerns were addressed adequately, we would appreciate an adjustment of their scores.

---

### Meta-Review · Area_Chair_UsWM · 2024-12-21

**Metareview:**

This paper proposes Chung-Lu Cooperative Mean Field Games (CLCMFG) for learning policies in large-scale sparse networks, introducing a two-systems approximation for handling varying node degrees. While showing promising empirical results on real networks, the work lacks theoretical foundations in multiple crucial aspects including convergence guarantees, solution properties analysis, and rigorous error bounds for the two-systems approximation.

**Additional Comments On Reviewer Discussion:**

Reviewers uniformly rated the paper marginally below acceptance (5/10), raising at least two key concerns:

- Theoretical Analysis: The absence of convergence guarantees and solution properties for the proposed algorithms remained a key concern. While the authors highlighted empirical validation, these theoretical gaps weaken the overall impact of the work.
- Comparison with Existing Methods: The proposed framework's advantages over existing approaches (e.g., graphon MFGs or multi-agent RL) were not convincingly established. Reviewers highlighted unclear distinctions and the need for better comparative analysis.

---

### Decision · Program_Chairs · 2025-01-22

Reject